# Hierarchical Visual Feature Aggregation for OCR-Free Document Understanding

**Jaeyoo Park** [1]    **Jin Young Choi**[2]    **Jeonghyung Park**[3]    **Bohyung Han**[1,2]

[1]ECE & [2]IPAI, Seoul National University
[3]Samsung SDS
{bellos1203, jychoi999, bhhan}@snu.ac.kr
jeong.h.park@samsung.com

## Abstract

We present a novel OCR-free document understanding framework based on pre-trained Multimodal Large Language Models (MLLMs). Our approach employs multi-scale visual features to effectively handle various font sizes within document images. To address the increasing costs of considering the multi-scale visual inputs for MLLMs, we propose the Hierarchical Visual Feature Aggregation (HVFA) module, designed to reduce the number of input tokens to LLMs. Leveraging a feature pyramid with cross-attentive pooling, our approach effectively manages the trade-off between information loss and efficiency without being affected by varying document image sizes. Furthermore, we introduce a novel instruction tuning task, which facilitates the model's text-reading capability by learning to predict the relative positions of input text, eventually minimizing the risk of truncated text caused by the limited capacity of LLMs. Comprehensive experiments validate the effectiveness of our approach, demonstrating superior performance in various document understanding tasks.

## 1  Introduction

In recent years, significant progress has been made in the development of Multimodal Large Language Models (MLLMs) trained on large-scale datasets [1–5], demonstrating remarkable performance in various tasks, such as visual question answering [6, 7], image captioning [8–10], image-text retrieval [11–13], visual grounding [14, 15] etc. MLLMs also exhibit the emergent capability achieving robust Optical Character Recognition (OCR) [9] performance by leveraging extensive pretraining data. Such advancements highlight the potential of MLLMs for practical and challenging tasks related to document understanding, by reducing reliance on external OCR engines which often struggle with complex layouts and varied fonts. Additionally, OCR-based models typically require additional modules to process explicit OCR outputs, adding to system complexity.

Despite their promising abilities, document understanding frameworks without external OCR engines [16–19] face significant challenges, particularly in considering multiple scales of document images, as text and visual elements often vary in size and style. Complex graphical elements, such as charts and diagrams, along with non-standard layouts, pose additional difficulties for the model due to limited receptive fields. This highlights the need for the frameworks to effectively handle the variability in document scales to achieve accurate and comprehensive understanding.

Our goal is to process visual documents over multiple scales rather than a single fixed scale. However, augmenting visual features for multiple scales inevitably increases costs, which is particularly critical for MLLMs, where LLMs have quadratic complexity due to the self-attention mechanism. To tackle this issue, we propose the Hierarchical Visual Feature Aggregation (HVFA) module to reduce the

38th Conference on Neural Information Processing Systems (NeurIPS 2024).

number of visual tokens by incorporating cross-attentive pooling within a feature pyramid. This design accommodates various document image sizes, effectively balancing the trade-off between information retention and efficiency. By reducing the visual features before feeding them into LLMs, our module maintains the original complexity of the LLMs.

In addition to incorporating multi-scale visual inputs, we introduce novel instruction tuning tasks to ensure the model's comprehensive text reading ability and capture layout information effectively. While [19] proposes reading all of the texts in images, we observed that certain documents contain more texts than LLMs can hold, resulting in information loss during training. To address this challenge, we propose an efficient and effective alternative task of reading, where the model learns to read partial texts in images based on their relative positions. Furthermore, teaching the model to predict the relative positions of given texts, which is an inverse task to the aforementioned reading task, enhances its ability to recognize overall layout information.

The main contributions of our paper are summarized as follows:

- We present a novel framework for OCR-free document understanding, built on a pretrained large-scale multi-modal foundation model, which integrates multi-scale visual features to handle varying font sizes in document images.

- We introduce the Hierarchical Visual Feature Aggregation (HVFA) module, which employs cross-attentive pooling to effectively balance information preservation and computational efficiency, addressing the escalating costs associated with detailed visual inputs.

- We employ a novel instruction tuning task, which aims to predict the relative positions of input text, to enhance the model's comprehensive text reading capability.

- The proposed approach presents remarkable performance gains on the multiple document understanding benchmarks among OCR-free models.

The rest of the paper is organized as follows. Section 2 reviews the related work. The details of our approach are described in Section 3, and the experimental results are presented in Section 4. Finally, we conclude this paper in Section 5.

## 2 Related Work

This section provides an overview of multimodal large language models and their applications to document understanding tasks. Research on document understanding typically falls into two categories: those that incorporate off-the-shelf OCR engines and those that bypass OCR extractors. From the extensive literature on this topic, we highlight the studies most relevant to our work.

### 2.1 Multimodal Large Language Models

Contrary to early multimodal learning approaches that focused on joint embedding learning for diverse downstream tasks [1–5, 20–24], recent frameworks [9, 12, 13, 25] have shifted toward generating open-ended language responses by leveraging LLMs [26, 27]. This evolution has contributed to various architectural innovations; GiT [9], BLIP [12], and Qwen-VL [28] pioneer encoder-decoder architectures for unified vision-language understanding and generation. Building upon these foundations, BLIP-2 [13] and mPLUG-Owl [25] introduce resamplers (Q-Formers) to bridge the vision-language modality gap through query-relevant feature extraction and dimensionality reduction. Further advances have come through InstructBLIP [29] and LLaVA [30], which enhance multimodal capabilities by instruction tuning on vision-language conversation data.

Recent studies explore MLLMs' text-reading capabilities and their relationship with image resolution. Wang et al. [9] shows MLLMs gradually learn to read text in images through large-scale pretraining. However, Liu et al. [31] reveals that most MLLMs are constrained by their visual encoder architecture designed only for 224×224 resolution inputs, significantly limiting their ability to capture fine-grained details in text-centric tasks. While Monkey [32] demonstrates enhanced performance by supporting high-resolution inputs, their analysis was limited to single-scale processing. These investigations collectively indicate that, although MLLMs can read text in images, their performance is significantly influenced by input resolutions, affecting their ability to handle diverse font sizes and capture fine-grained textual details.

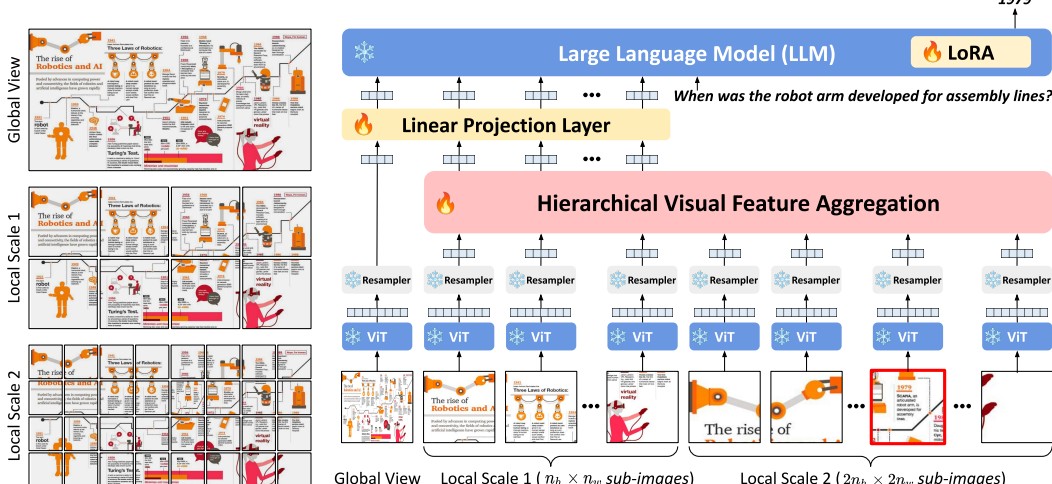

Figure 1: Illustration of the proposed framework. Our model adopts visual features from multiple scales, which are aggregated through the Hierarchical Visual Feature Aggregation (HVFA) module. The aggregated features are then fed into an LLM to generate language response in an autoregressive manner. The sub-image highlighted by the red box contains the text relevant to an input question, which requires accurately recognizing visually detailed elements from high resolution image.

## 2.2 OCR-based Document Understanding Models

Early studies adopted external OCR engines to explicitly model and understand the texts within images [33–39]. LayoutLM [33] directly extends BERT [40] to jointly model interactions between text, visual and layout information for the first time. Based on LayoutLM, there have been efforts to introduce various pretraining objectives to learn layout information effectively by incorporating image-text alignment [34], cell-level information from OCR outputs [36], and multi-modal masking [35, 37]. Despite their strong performance on document understanding benchmarks, these models rely on off-the-shelf OCR engines and struggle with challenging cases such as diverse fonts, handwritten texts, and degraded historical documents. Moreover, they require additional modules for processing OCR results, which increases computational complexity.

## 2.3 OCR-free Document Understanding Models

To address the drawbacks of OCR-based models, OCR-free models powered by end-to-end multi-modal pretraining have been proposed. Dessurt [16] and Donut [17] combine a learnable visual encoder with an autoregressive text decoder for text recognition. Pix2Struct [18] focuses on layout understanding by learning from HTML data and masked webpage screenshots. However, these methods suffer from high computational costs because they require pretraining models from scratch before task-specific fine-tuning. UReader [19] takes a different approach by leveraging pretrained MLLMs, reducing training costs while handling diverse document types through document instruction tuning and shape-adaptive cropping. Despite these advances, current OCR-free models still struggle to capture diverse visual scales and font sizes, often missing local details in documents.

## 3 Method

This section describes our main idea, including multi-scale visual inputs, the Hierarchical Visual Feature Aggregation (HVFA) module, and relative text-position prediction tasks in detail.

### 3.1 Overview

MLLMs generate instruction-following responses given multi-modal inputs, an image $I$ and text (instruction) $T$. The output token sequence with a length of $n_s$ is denoted by $S = \{s_1, ..., s_{n_s}\}$, where the probability of generating $s_i$ is given by $p(s_i|I, T, S_{<i})$, $1 \leq i \leq n_s$. Given the correct

answer $Y = \{\mathbf{y}_1, ..., \mathbf{y}_{n_y} | \mathbf{y}_i \in \mathbb{R}^{|V|}\}$, where $n_y$ is the length of the answer and $V$ is the vocabulary set, the learning objective of the MLLMs, parameterized by $\theta$, is given by

$$\theta^* = \arg\min_{\theta} \ \mathcal{L}_{\text{MLLM}}(I, T; \theta)$$

$$= \arg\min_{\theta} \ \frac{1}{|\mathcal{D}|} \sum_{(I,T)\in\mathcal{D}} \left[ -\sum_{i=1}^{n_s} \sum_{v=1}^{|V|} \mathbf{y}_{i,v} \log p(s_i = v | I, T, S_{<i}; \theta) \right], \tag{1}$$

where $\mathcal{D}$ is the image-text dataset.

MLLMs generally comprise three modules: 1) a vision encoder, 2) a projector, and 3) a large language model (LLM). First, we generate visual inputs at multiple scales from a given document image. The vision encoder extracts visual features for detailed image understanding. The resampler-based projector converts these visual features into a compact set of query tokens for the language model. Unlike traditional MLLMs, which feed these query tokens directly into the LLM, we introduce the Hierarchical Visual Feature Aggregation module to reduce the number of visual query tokens while preserving local information. Finally, the LLM processes both the visual and instruction tokens to generate a response autoregressively. Figure 1 illustrates the overall framework.

### 3.2   Multi-Scale Visual Features

We focus on developing a document understanding framework that incorporates multi-scale visual inputs to address various image scales for MLLMs. However, using pretrained MLLMs constrains the input resolution to a small, square-shaped configuration defined by the visual encoder. This limitation hampers capturing detailed information in high-resolution images and leads to spatial distortions when handling document images of diverse resolutions and aspect ratios.

We adopt shape-adaptive cropping (SAC) [19] to generate multiple sub-images, allowing our model to handle varying aspect ratios and resolutions. We split each image using a predefined grid $\{g = (n_h \times n_w) | n_h, n_w \in \mathbb{N}\}$, where $n_h$ and $n_w$ respectively denote the number of rows and columns for the grid $g$. Then, each sub-image is resized to fit the visual encoder. Note that the optimal grid $g^* = (n_h^* \times n_w^*)$ for each image is selected by two metrics; resolution coherence, and shape similarity. Refer to Appendix C for more details.

Although SAC handles diverse aspect ratios and high-resolution images, it may miss local details within each sub-image due to its receptive fields of fixed-size. To address this issue, we consider an additional level of detail by applying $2\times$ upscaling to each sub-image to capture detailed visual features and small text fonts through enlarged local resolution. To be specific, this augmentation process involves subdividing each sub-image into half along each spatial dimension, resulting in $2n_h \times 2n_w$ sub-images. We again resize the subdivided images to ensure compatibility with the visual encoder. By incorporating these high-resolution representations, the model now observes the input image in three different levels: global, $n_h \times n_w$, and $2n_h \times 2n_w$ scales.

Unlike vision encoders and resampler-based projectors, which process sub-images in batches and thereby exhibit linear computational complexity, LLMs demonstrate quadratic complexity with respect to the number of sub-images. Thus, when the number of visual inputs increases from $n$ to $5n$, the computational burden of LLMs escalates 25 times, while that of vision encoders and resamplers grows 5 times. Because LLMs are represented by networks with significantly more channels than other models, this quadratic increase of computational cost is problematic in many practice scenarios.

### 3.3   Hierarchical Visual Feature Aggregation

The HVFA module leverages a feature pyramid to fuse multi-scale features for LLMs and reduces computational overhead. We first construct a feature pyramid using the visual features from two adjacent scales, $i^{\text{th}}$ and $(i+1)^{\text{st}}$ scales. Cross-attentive pooling is then applied to features at scale $(i+1)$ to compress and preserve detailed information, and the pooled features are aggregated with low-resolution features at scale $i$. Figure 2 depicts the mechanism of the HVFA module.

Let $\mathbf{F}_i \in \mathbb{R}^{n_h^i \times n_w^i \times q \times c}$ be the visual features for the $i^{\text{th}}$ scale extracted from resampler, where $(n_h^i \times n_w^i)$ is the number of sub-images at $i^{\text{th}}$ scale, $q$ denotes the number of queries from resampler, and $c$ is the number of channels. We perform $2 \times 2$ max-pooling on $\mathbf{F}_{i+1} \in \mathbb{R}^{n_h^{i+1} \times n_w^{i+1} \times q \times c}$ to

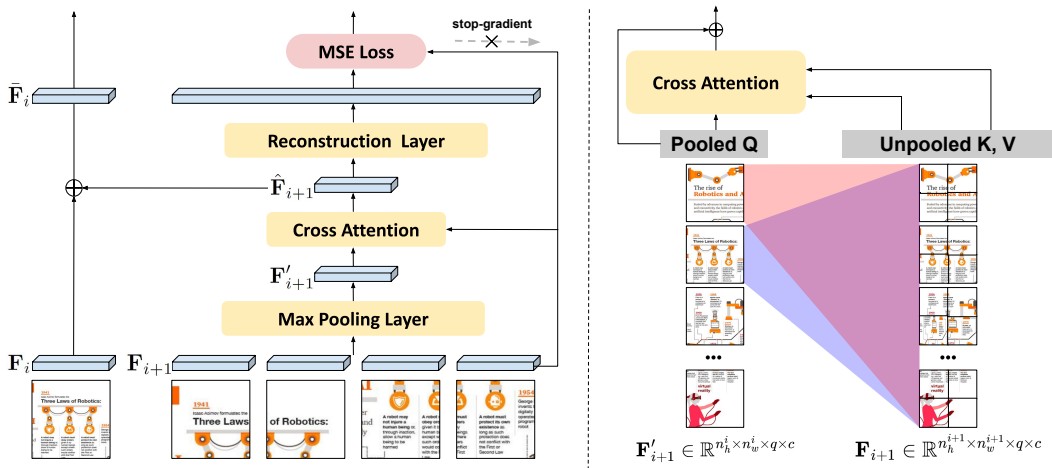

Figure 2: Illustration of the Hierarchical Visual Feature Aggregation (HVFA) module. (Left) HVFA aggregates high-resolution visual features to low-resolution features leveraging feature pyramid structure. (Right) In cross-attentive pooling, each sub-image-feature attends to all of the fine-grained visual features, compressing and preserving more detailed information.

obtain a pooled feature $\mathbf{F}'_{i+1} \in \mathbb{R}^{n_h^i \times n_w^i \times q \times c}$. To reduce the information loss by the max-pooling operation, we conduct the following operations including a lightweight cross-attention:

$$\hat{\mathbf{F}}_{i+1} = \mathbf{F}'_{i+1} + \text{Softmax}\left(\frac{(\mathbf{F}'_{i+1}\mathbf{W}_{\text{query}})(\mathbf{F}_{i+1}\mathbf{W}_{\text{key}})^T}{\sqrt{d}}\right)(\mathbf{F}_{i+1}\mathbf{W}_{\text{value}})\mathbf{W}_{\text{proj}} \qquad (2)$$

$$\bar{\mathbf{F}}_i = \mathbf{F}_i + \hat{\mathbf{F}}_{i+1}, \qquad (3)$$

where $\mathbf{W}_{\text{query}}, \mathbf{W}_{\text{key}}, \mathbf{W}_{\text{value}} \in \mathbb{R}^{c \times d}$, and $\mathbf{W}_{\text{proj}} \in \mathbb{R}^{d \times c}$ are learnable parameters, and $d$ is the embedding dimension. In Equation (3), a pooled feature $\mathbf{F}'_{i+1}$ corresponds to a query and the original feature, $\mathbf{F}_{i+1}$, works as a key and a value. The layer normalization is omitted for simplicity of the expression. We feed $\bar{\mathbf{F}}_i$ to LLMs to generate the language responses. Note that our method maintains the original complexity of LLMs, despite considering multiple scales in practice, because the feature pyramid structure aggregates them into the initial local scale of visual features generated by SAC.

We assume that well-compressed features retain essential information necessary for the reconstruction of the original features. After obtaining the pooled features, based on this assumption, we introduce a small decoder network, used only for training, to reconstruct the original features from the pooled features. The decoder is trained with the mean squared error (MSE) loss between the reconstructed and original features, which is given by

$$\mathcal{L}_{\text{MSE}}(\hat{\mathbf{F}}_{i+1}, \mathbf{F}_{i+1}) = \mathbb{E}[(r(\hat{\mathbf{F}}_{i+1}) - \text{stopgrad}(\mathbf{F}_{i+1}))^2], \qquad (4)$$

where $r(\cdot)$ is an MLP-based reconstruction network and $\text{stopgrad}(\cdot)$ indicates a stop-gradient operation preventing the collapse in training. By the MSE loss, we guide the model to retain important visual details while reducing feature dimensionality. The final objective function is given by

$$\mathcal{L}_{\text{Final}} = \mathcal{L}_{\text{MLLM}} + \lambda \mathcal{L}_{\text{MSE}}, \qquad (5)$$

where $\lambda$ is the weight for the balance between two terms.

Our approach is similar to pooling-based methods such as MViT [41] and MViT-v2 [42], which construct attention mechanisms utilizing pooled features as keys and values to build efficient ViT variants [43]. However, unlike these methods, we use pooled features as queries to reduce query size rather than optimizing attention mechanisms themselves.

Our cross-attentive pooling can be seen as a form of resampler [13, 44] that reduces spatial dimensionality. In this interpretation, the pooled features, $\mathbf{F}'_{i+1}$, act as the query tokens for learning. Unlike traditional query-based resamplers that require a new set of trainable query vectors, our method uses pooled features, which are more effective for resampling than randomly initialized query vectors, as discussed further in Section 4.4.

Table 1: Examples for Relative Text-Position Prediction Task. Refer to Appendix I for the full list.

| Task | Instruction Templates |
|------|----------------------|
| RPT (first) | Human: What's in the first 30% of the image text? AI: {corresponding texts}. 
 Human: Identify words from the first 15% of the image text. AI: {corresponding texts}. |
| RPT (middle) | Human: What are the words located between 10% and 55% of the text in the image? AI: {corresponding texts}. 
 Human: List words found between 20% and 40% in the image text. AI: {corresponding texts}. |
| RPT (last) | Human: Identify words from the last 16% of the image text. AI: {corresponding texts}. 
 Human: Which words make up the last 40% of the text in the image? AI: {corresponding texts}. |
| PTP | Human: Specify the relative position within the image where {query texts} is found. AI: 15%-30%. 
 Human: Where is the text {query texts} located within the image? AI: 40%-80%. |

## 3.4 Relative Text-Position Prediction Task

Another key challenge in document understanding is learning to read text using layout information. Although [19] trains models to read entire document text, this approach entails significant computational challenges and information losses due to text truncation caused by the input capacity of LLMs. To address these limitations, we introduce two novel tasks: Reading Partial Text (RPT) and Predicting Text Position (PTP). RPT focuses on reading specific text segments at given positions, while PTP aims to predict the positions of given text segments. These tasks facilitate enhanced layout understanding with reduced computational cost.

Suppose that we can access full document text, represented by a sequence of tokens and we adopt a standard reading order of text (top-to-bottom, left-to-right). For the RPT task, we randomly select one of three types: 1) "*first*": reading up to $p_{end}\%$ of the text, 2) "*middle*": reading from $p_{start}\%$ to $p_{end}\%$ of the text, and 3) "*last*": reading from $p_{start}\%$ to the end. We also sample an instruction template for the selected type, where the instruction template examples are presented in Table 1 and the full list is available in Appendix I. We then compute the maximum text coverage, $c_{\max}$, to mitigate text truncation, which is given by

$$c_{\max} = \min(1.0, l_{\max}/l), \tag{6}$$

where $l$ is the length of the tokenized text and $l_{\max}$ is the maximum sequence length acceptable by LLMs.

We set a threshold for the minimum coverage as a hyperparameter, $c_{\min}$, to ensure that sufficient text is read. The range for reading, denoted by $t_{\text{range}}$, is given by

$$t_{\text{range}} = \begin{cases} c_{\max} & \text{if } c_{\max} \leq c_{\min} \\ \text{random}(c_{\min}, c_{\max}) & \text{otherwise,} \end{cases} \tag{7}$$

where $\text{random}(a, b)$ denotes a uniform sampling from an interval $(a, b)$. Then, the positions to predict are expressed as follows:

$$(p_{\text{start}}, p_{\text{end}}) = \begin{cases} (0, \ t_{\text{range}}) & \text{if type} = \text{'first'} \\ (p'_{\text{start}}, \ p'_{\text{start}} + t_{\text{range}}) & \text{else if type} = \text{'middle'} \\ (100 - t_{\text{range}}, 100) & \text{else if type} = \text{'last'.} \end{cases} \tag{8}$$

where $p'_{start} = \text{random}(0, 100 - t_{\text{range}})$.

For the PTP task, given a segment of text, the model aims to infer its relative position within the whole text. We determine text segments and their relative positions using the same process as in the RPT task. Through diverse training examples, the model learns to understand spatial relationships between text segments and other components in a document.

## 4 Experiments

### 4.1 Datasets and Evaluation Metrics

We utilize document instruction dataset corpora for training, following [19]. The training dataset includes various types of document images, including tables, charts, natural images, and web

Table 2: Configurations of OCR-free document understanding baselines.

| Method | # Model Params | # Trainable Params | # Pretraining Data | # Fine-tuning data |
|---|---|---|---|---|
| *Document-specific Pretraining* | | | | |
| Dessurt [16] | 127M | 127M | 11M | Task-specific |
| Donut [17] | 143M | 143M | 13M | Task-specific |
| Pix2Struct$_{Base}$ [18] | 282M | 282M | 80M | Task-specific |
| Pix2Struct$_{Large}$ [18] | 1.3B | 1.3B | 80M | Task-specific |
| *MLLM-based Instruction Tuning* | | | | |
| Qwen-VL [28] | 9.6B | 207M | 1.4B+76.8M | Task-specific |
| Monkey [32] | 9.8B | 207M | 1.4B+76.8M | 1.44M |
| BLIP-2-OPT-2.7B + UReader [19] | 3.8B | 8M | 129M | 650K |
| BLIP-2-OPT-2.7B + Ours | 3.8B | 14M | 129M | 650K |
| mPLUG-Owl-7B + UReader [19] | 7.2B | 86M | 1.1B | 650K |
| mPLUG-Owl-7B + Ours | 7.2B | 96M | 1.1B | 650K |

Table 3: Performance comparison with other OCR-free document understanding baselines. The bold-faced numbers indicate the best performance in each column.

| Method | DocVQA | InfoVQA | DeepForm | KLC | WTQ | TabFact | ChartQA | VisualMRC | TextVQA | TextCaps |
|---|---|---|---|---|---|---|---|---|---|---|
| *Document-specific Pretraining* | | | | | | | | | | |
| Dessurt [16] | 63.2 | – | – | – | – | – | – | – | – | – |
| Donut [17] | 67.5 | 11.6 | 61.6 | 30.0 | 18.8 | 54.6 | 41.8 | 93.91 | 43.5 | 74.4 |
| Pix2Struct$_{Base}$ [18] | 72.1 | 38.2 | – | – | – | – | 56.0 | – | – | 88.0 |
| Pix2Struct$_{Large}$ [18] | 76.6 | 40.0 | – | – | – | – | 58.6 | – | – | 95.5 |
| *MLLM-based Instruction Tuning* | | | | | | | | | | |
| Qwen-VL [28] | 65.1 | 35.4 | 4.1 | 15.9 | 21.6 | – | **65.7** | – | 63.8 | – |
| Monkey [32] | 66.5 | 36.1 | 40.6 | 32.8 | 25.3 | – | 65.1 | – | **67.6** | – |
| BLIP-2-OPT-2.7B + UReader [19] | 38.7 | 22.9 | 5.6 | 18.3 | 17.4 | 58.5 | 37.1 | 214.3 | 43.2 | 126.3 |
| BLIP-2-OPT-2.7B + Ours | 51.4 | 29.6 | 14.6 | 23.8 | 21.2 | 59.9 | 50.4 | **228.7** | 57.3 | **135.2** |
| mPLUG-Owl-7B + UReader [19] | 65.4 | 42.2 | 49.5 | 32.8 | 29.4 | 67.6 | 59.3 | 221.7 | 57.6 | 118.4 |
| mPLUG-Owl-7B + Ours | **72.7** | **45.9** | **53.0** | **36.7** | **34.5** | **68.2** | 63.3 | 226.4 | 59.2 | 123.1 |

page screenshots. The datasets used are DocVQA [45], InfographicsVQA [46], DeepForm [47], KleisterCharity [48], WikiTableQuestions [49], TabFact [50], ChartQA [51], VisualMRC [52], TextVQA [53], and TextCaps [54]. The combined dataset comprises approximately 650K image-instruction pairs. For more details about the datasets, please refer to Appendix E.

We evaluate our model on the test splits of each dataset. For DocVQA and InfoVQA, we employ the ANLS score [55], while we adopt the F1 score for DeepForm and KLC. The performance on WTQ, TabFact, and TextVQA is measured using accuracy, whereas ChartQA utilizes relaxed accuracy [56]. The metric for TextCaps and VisualMRC is the CIDEr score [57]. Evaluations for TextVQA and TextCaps are conducted on their respective official challenge websites.

## 4.2 Implementation Details

We conduct experiments on two pretrained resampler-based MLLMs: BLIP-2-OPT-2.7B [13] and mPLUG-Owl-7B [25]. We utilize LoRA [58] to fine-tune the language model efficiently while keeping the vision encoder frozen. For the BLIP-2-based model, we freeze the resampler and apply LoRA, whereas for the mPLUG-Owl-based model, we train the resampler to ensure a fair comparison with [19]. Regarding the SAC, we set the maximum number of sub-images to 9 for the BLIP-2-based model and 20 for the mPLUG-Owl-based model. Refer to Appendix D for more details.

## 4.3 Quantitative Results

We compare our framework to other OCR-free baselines, including document-specific pretraining approaches [16–18] for completeness. Given that previous methods utilize varying numbers of model parameters, trainable parameters, and training data, we provide the detailed configurations in Table 2. It is important to note that our comparisons are made with OCR-free methods based on comparable MLLMs in terms of model size and dataset scale. Table 3 presents the overall results of the proposed algorithm across 10 document understanding benchmarks. When integrated with both BLIP-2 and mPLUG-Owl, our approach consistently outperforms UReader [19] by a significant margin across all benchmarks. Among MLLM-based methods, our framework outperforms Qwen-

Table 4: Results of main ablation studies with the BLIP-2-based model. Note that the reconstruction loss (Recon) only comes with MS + HVFA.

| MS + HVFA | Recon | RTPP | DocVQA | InfoVQA | DeepForm | KLC | WTQ | TabFact | ChartQA | VisualMRC | TextVQA | TextCaps |
|---|---|---|---|---|---|---|---|---|---|---|---|---|
| | | | 35.17 | 20.34 | 4.14 | 18.50 | 15.36 | 53.37 | 31.76 | 201.39 | 43.20 | 126.34 |
| √ | | | 45.22 | 25.87 | 8.23 | 19.82 | 18.14 | 58.28 | 44.16 | 221.38 | 48.56 | 129.98 |
| | | √ | 40.18 | 25.19 | 6.49 | 18.97 | 17.80 | 56.85 | 38.88 | 219.74 | 47.54 | 128.39 |
| √ | √ | | 47.36 | 27.37 | 10.92 | 19.17 | 18.24 | 58.14 | 46.70 | 222.56 | 52.65 | 130.86 |
| √ | | √ | 50.46 | 28.48 | 13.18 | **23.82** | 20.06 | 59.83 | 49.48 | 226.65 | 56.04 | 134.06 |
| √ | √ | √ | **51.38** | **29.60** | **14.58** | 23.78 | **21.15** | **59.87** | **50.41** | **228.65** | **57.32** | **135.24** |

Table 5: Ablation study on the variations of hierarchical visual feature aggregation techniques. We employ a BLIP-2-based model for experiments.

| Variations | DocVQA | InfoVQA | DeepForm | KLC | WTQ | TabFact | ChartQA | VisualMRC | TextVQA | TextCaps |
|---|---|---|---|---|---|---|---|---|---|---|
| (a) *Spatial Dimension Reduction* | | | | | | | | | | |
| Max Pooling-only | 43.23 | 25.44 | 7.88 | 19.32 | 18.14 | 58.82 | 44.36 | 222.49 | 45.91 | 126.34 |
| Linear Projectors | 46.11 | 26.44 | 10.62 | 21.32 | 18.74 | 58.52 | 46.62 | 224.74 | 48.27 | 129.68 |
| Cross-Attentive Pooling (Ours) | **51.38** | **29.60** | **14.58** | **23.78** | 21.15 | **59.87** | **50.41** | 228.65 | **57.32** | **135.24** |
| Cross-Local-Attentive Pooling | 50.32 | 28.60 | 14.38 | 23.56 | **22.15** | 59.66 | 50.22 | **228.82** | 56.78 | 134.34 |
| (b) *Query Token Initialization* | | | | | | | | | | |
| Random Vectors | 48.07 | 27.34 | 11.40 | 21.82 | 19.85 | 58.42 | 48.01 | 223.37 | 53.48 | 130.22 |
| Max-Pooled Features (Ours) | **51.38** | **29.60** | **14.58** | **23.78** | **21.15** | **59.87** | **50.41** | **228.65** | **57.32** | **135.24** |
| (c) *Stop-Gradient* | | | | | | | | | | |
| w/o stopgrad | 49.65 | 26.80 | 12.16 | 19.73 | 19.14 | 59.11 | 48.85 | 223.42 | 55.24 | 128.44 |
| w/ stopgrad (Ours) | **51.38** | **29.60** | **14.58** | **23.78** | **21.15** | **59.87** | **50.41** | **228.65** | **57.32** | **135.24** |
| (d) *Reconstruction Loss* | | | | | | | | | | |
| $\lambda = 1.0$ | 49.90 | 28.51 | 14.41 | 22.75 | 20.13 | 59.03 | 49.76 | 225.51 | 56.79 | 133.03 |
| $\lambda = 0.1$ (Ours) | 51.38 | **29.60** | **14.58** | 23.78 | **21.15** | **59.87** | **50.41** | **228.65** | **57.32** | **135.24** |
| $\lambda = 0.01$ | **51.84** | 28.14 | 13.84 | **23.87** | 20.39 | 59.54 | 50.05 | 227.57 | 57.03 | 135.17 |
| $\lambda = 0$ | 50.46 | 28.48 | 13.18 | 23.82 | 20.06 | 59.83 | 49.48 | 226.65 | 56.04 | 134.06 |

VL-based methods [28, 32] in most benchmarks, despite using a smaller set of pretraining data and smaller model. The results show that our model successfully transfer the knowledge of pretrained MLLMs to understand document samples.

## 4.4 Ablation Study

We perform several ablation studies with the BLIP-2-based model to validate the effectiveness of each component of our approach.

**Effect of each component** Based on the results in Table 4, we observe several key findings. First, our multi-scale hierarchical visual feature aggregation (MS + HVFA) significantly improves performance, by leveraging the benefits of multi-scale features. Second, incorporating a reconstruction layer to learn how to reconstruct (Recon) the original features from the pooled features aids in the information compression process, in most cases. Finally, the Relative Text-Position Prediction Task (RTPP) enhances overall performance by improving the model's text-reading ability.

**Spatial dimensionality reduction methods** Table 5(a) compares different operations for reducing the spatial dimensions of finer-scale features within the HVFA module. Our cross-attentive pooling outperforms simple max-pooling, which suffers from significant information loss due to its non-learnable nature. We also compare our method against a stack of linear projection layers with the same parameter count. The results demonstrate that cross-attentive pooling achieves superior performance by learning more effective feature relationships through its attention mechanism, enabling context-aware information aggregation. Additionally, we investigate a local-attention-based cross-attention mechanism, where the cross-attention is restricted to each sub-image and its corresponding scaled-up features within the feature pyramid. Its slightly inferior results indicate that the pooling operation needs to aggregate information from various regions to be effective.

**Query token initialization strategy** Our cross-attentive pooling can be regarded as a special form of resampler since it refines high-resolution features while reducing spatial dimensionality. In this

Table 6: Ablation study on text reading tasks. We conduct our ablation study with the BLIP-2-based model. The bold-faced numbers indicate the best performance in each column.

| RFT [19] | RPT | PTP | DocVQA | InfoVQA | DeepForm | KLC | WTQ | TabFact | ChartQA | VisualMRC | TextVQA | TextCaps |
|---|---|---|---|---|---|---|---|---|---|---|---|---|
| | | | 47.36 | 27.37 | 10.92 | 19.17 | 18.24 | 56.14 | 43.88 | 222.56 | 52.65 | 130.86 |
| √ | | | 49.23 | 28.16 | 12.58 | 21.29 | 19.42 | 58.46 | 46.05 | 224.39 | 55.78 | 133.55 |
| √ | | √ | 50.62 | 29.49 | 12.55 | 22.88 | 20.51 | 59.28 | 48.41 | 226.55 | 57.19 | 134.44 |
| | √ | | 50.46 | 28.44 | **14.88** | 22.55 | 20.88 | 59.12 | 48.28 | 226.56 | 56.91 | 134.92 |
| | √ | √ | **51.38** | **29.60** | 14.58 | **23.78** | **21.15** | **59.87** | **50.41** | **228.65** | **57.32** | **135.14** |

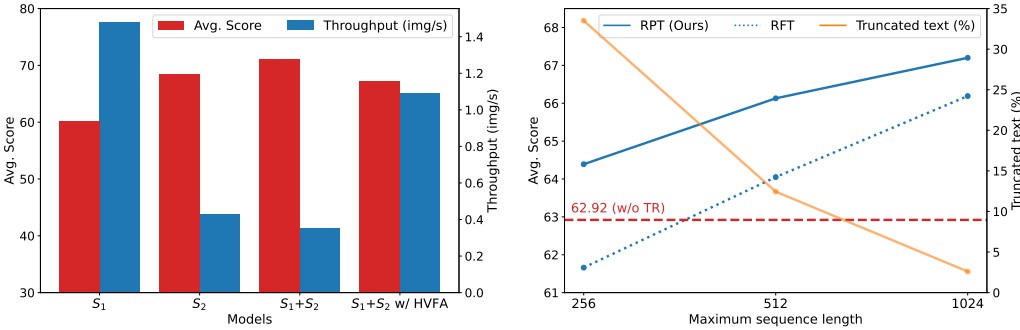

Figure 3: Performance analysis on visual and textual inputs. (Left) Impact of visual input scale on model performance. We compare four variants of our model: with the first scale ($S_1$), with the second scale ($S_2$), with multiple scales ($S_1 + S_2$), and with multiple scales with HVFA ($S_1 + S_2$ w/ HVFA, ours). (Right) Impact of truncated text in the text reading task on model performance by varying sequence length capacity of LLM.

context, the pooled features, denoted as $\mathbf{F}'_{i+1}$, serve as query tokens [13, 44] for the learning process. To explore this perspective, we compare our approach with randomly initialized queries of the same size as $\mathbf{F}'_{i+1}$. As demonstrated in Table 5(b), using max-pooled features for query initialization yields superior results, even without tintroducing additional parameters for training.

**Role of stop-gradient** Table 5(c) presents the role of the "stop-gradient" operation in Equation (4), where the architectures and all hyperparameters remain unchanged. Empirically, $\mathcal{L}_{\text{MSE}}$ drops to zero at an early iteration without the stop-gradient operation due to the convergence at a trivial solution; it leads to a significant degradation in performance.

**Impact of $\lambda$ to balance the loss** We tested several different values of $\lambda$ to determine the optimal balance for the final loss function, $\mathcal{L}_{\text{Final}}$, in Equation (5). Results in Table 5(d) indicate that $\lambda = 0.1$ achieves the best performance. A value of $\lambda = 0.01$ also improves performance compared to $\lambda = 0$, whereas $\lambda = 1.0$ yields worse results than $\lambda = 0$ on some benchmarks. These findings suggest that, while $\mathcal{L}_{\text{MSE}}$ positively contributes by learning compressed features, excessively strong compression hampers the model's primary goal, extracting text-related features.

**Effectiveness of relative text-position prediction task** Table 6 illustrates the effectiveness of RPT compared to RFT, implying that concentrating on key text segments guided by their relative positions enhances document understanding. Notably, RFT may lead to information loss due to text truncation, given the limited capacity of LLMs. Furthermore, PTP improves the performance of both RPT and RFT tasks, showing that predicting the relative positions of partial text strengthens the model's comprehension of document layout and content. This highlights the importance of considering text position in document understanding tasks.

## 4.5 Complexity Analysis on Visual Input Scale

To analyze the trade-off between performance and computational cost, we compare variants of our model that process different scales of visual inputs: the first scale ($S_1$), the second scale ($S_2$), multiple scales ($S_1 + S_2$), and multiple scales with HVFA ($S_1 + S_2$ w/ HVFA, ours). Figure 3(Left) presents the average benchmark scores and throughput for each model. The throughput is measured on a single NVIDIA Quadro RTX GPU using the largest possible batch size for each model. where the input size

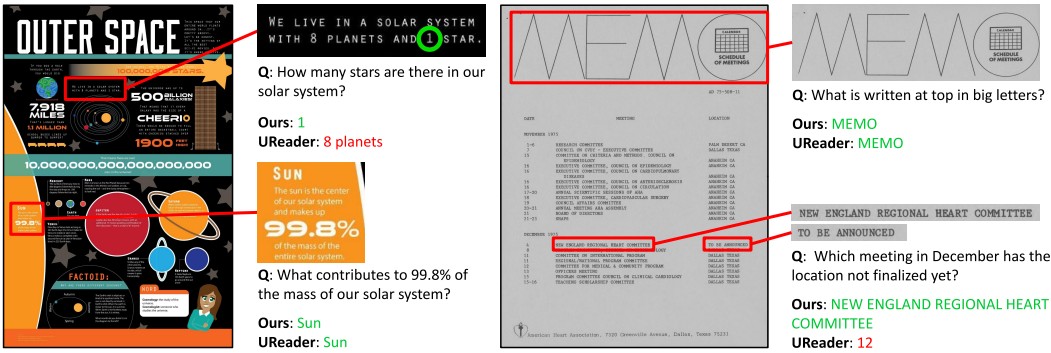

Figure 4: Our method vs. UReader [19] on DocVQA [45] and InfographicVQA [46].

of the visual encoder is $224 \times 224$. The results indicate that incorporating a finer scale significantly improves performance, albeit with increased computational cost ($S_1$ vs. $S_2$). Additionally, combining features from multiple scales boosts performance over single-scale features, though at the expense of higher computational load ($S_1$, $S_2$ vs. $S_1 + S_2$). Finally, our method efficiently integrates finer-scale features into a coarser scale, achieving a balance between performance gains and computational demands ($S_1 + S_2$ vs. $S_1 + S_2$ w/ HVFA).

## 4.6 Analysis on Text Reading Task

To demonstrate the effectiveness and robustness of RPT against truncation, we conducted experiments varying the maximum sequence length of LLMs during training. Figure 3(Right) presents the average benchmark scores and the proportion of truncated text data across the entire dataset. RPT consistently outperforms RFT in all settings, indicating that RPT is robust to the capacity constraints of LLMs. The design of RPT is helpful for mitigating truncation issues while RFT suffers from truncation of text reading data. Notably, with a maximum sequence length of 256, approximately 33.5% of the dataset consists of truncated texts, which is a considerable amount. Consequently, model performance declines even further compared to training without any text-reading task, as represented by the red dotted line in Figure 3(Right). Our approach ensures reliable data quality while incorporating stochasticity and positional information, contributing to its effectiveness. These results imply that our partial text reading task can inspire further research on knowledge distillation and the development of compact models for OCR-related tasks.

## 4.7 Visualization of Results

Figure 4 illustrates question-answer pairs from the DocVQA [45] and InfographicVQA [46] datasets for comparisons between our framework and UReader [19]. While both models can read large text, our model is more powerful in identifying small text than UReader. This highlights our model's strength in effectively handling text at multiple scales. Furthermore, our method delivers more accurate answers for questions requiring precise reading comprehension. For instance, to answer the question "Which meeting in December has the location not finalized yet?", the model needs to accurately read the text "To be announced" as demonstrated in the rightmost example in Figure 4.

## 5 Conclusion

We present a novel OCR-free document understanding framework that leverages a pretrained large-scale multi-modal foundation model. Our approach integrates multi-scale visual features to accommodate diverse font sizes within document images. To manage the increasing costs for detailed visual inputs, we propose the Hierarchical Visual Feature Aggregation (HVFA) module with cross-attentive pooling, effectively balancing information retention and computational efficiency while adapting to varying document sizes. Moreover, our framework introduces a novel instruction tuning task to enhance the model's text-reading ability by incorporating positional information within images. Extensive experimentation demonstrates the efficacy of our framework, achieving strong performance across a range of document understanding tasks.

## Acknowledgement

This work was partly supported by Samsung SDS, and the Institute of Information & communications Technology Planning & Evaluation (IITP) grants [No.RS-2022-II220959 (No.2022-0-00959), (Part 2) Few-Shot Learning of Causal Inference in Vision and Language for Decision Making, No.RS-2021-II211343, Artificial Intelligence Graduate School Program (Seoul National University), No.RS-2021-II212068, Artificial Intelligence Innovation Hub], and the National Research Foundation of Korea (NRF) grant [No.RS-2022-NR070855, Trustworthy Artificial Intelligence], funded by the Korean government (MSIT).

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

## A  Broader Impacts

Research on document understanding and large multimodal language models (MLLMs) can significantly impact various societal domains. In education, they enable personalized learning and greater accessibility to information, while in healthcare, they streamline medical documentation and enhance patient education. Legal and regulatory fields benefit from improved document review and compliance, and businesses gain through advanced customer service and financial analysis tools. However, addressing biases, ensuring privacy, and managing employment impacts are crucial ethical considerations. Moreover, vision system that can understand documents can be used in extracting privacy information from documents easily. Balancing these advancements with societal responsibilities can maximize the benefits for a more informed and equitable society.

## B  Limitations

Our work has some drawbacks inherited from LLMs, since our work directly employs pretrained LLMs. First, our work is that it is vulnerable to the bias in LLMs, which can cause misinterpreting the crucial and confidential documents. Next, given some document formats, the model may cause the privacy concerns. Our model has been trained only with the documents written in English and thus may struggle the documents written in other languages. While our work employs multi-scale visual features, we still constrains from using resampler-based MLLMs, and thus cannot fully exploit the characteristics of vision encoder, ViT. We believe that our method can directly be adopted to resampler-free pretrained models, since the proposed approach is simple.

## C  Shape-Adaptive Cropping

Shape-adaptive cropping (SAC) [19] is an image preprocessing method that generates multiple crops to handle varying aspect ratios and resolutions, enhancing versatility and performance. Given an image $I$ with dimensions $H \times W$, we split the image with one of the predefined grids $\{g = (n_h \times n_w)|n_h, n_w \in \mathbb{N}\}$, where $n_h, n_w$ denote the number of rows and columns of the grid $g$. Each image is then resized to $(n_h H_v, n_w W_v)$ to generate $g$ sub-images, where $H_v$, and $W_v$ are the spatial configuration defined by the pretrained vision encoder.

The suitable grid $g^* = (n_h^* \times n_w^*)$ for each image is selected based on two criteria; resolution coherence, and shape similarity. It's essential for the grid to maintain the image's resolution to the greatest extent possible, and the grid should conform to the aspect ratio of the input image. Given the predefined set of grids, we denote the resolution-related and resolution-agnostic intersection over union, as $s_{rr}$ and $s_{ra}$, respectively. $s_{rr}$ and $s_{ra}$ are defined as follows:

$$s_{rr}(I, g) = \text{IoU}((H, W), (n_h H_v, n_w W_v)) \tag{9}$$

$$s_{ra}(I, g) = \text{IoU}((\frac{n_w H}{W}, n_w), (n_h, n_w)). \tag{10}$$

Then the optimal grid $g^*$ is selected by:

$$g^* = \text{argmax}_g(s_{rr}(I, g) + s_{ra}(I, g)). \tag{11}$$

## D  Implementation Details

We experiment with two pretrained resampler-based MLLMs: BLIP-2-OPT-2.7B [13] and mPLUG-Owl-7B [25]. We utilize LoRA [58] to fine-tune the language model efficiently while keeping the vision encoder frozen. In the case of the resampler, we freeze it and apply LoRA for the BLIP-2-based model, while training it for mPLUG-Owl to ensure a fair comparison to UReader [19]. For LoRA, we set rank $r = 8$, and $\alpha = 32$. The maximum sequence length of the LLM is set to 2048 for all experiments, except for the one described in Section 4.6. Regarding the shape-adaptive cropping module, we set the maximum number of sub-images to 9 for the BLIP-2-based model and 20 for the mPLUG-Owl-based model. We employ the transformers library from huggingface[1] to constuct

---

[1] `https://huggingface.co/Salesforce/blip2-opt-2.7b`

BLIP-2-based model. For mPLUG-Owl, we implement our algorithm based on the official repository of UReader.[2]

For the Hierarchical Visual Feature Aggregation (HVFA) module, we employed a two-layer multi-head cross-attention layer with $d = 256$ and 12 heads. For $\mathcal{L}_{MSE}$, we utilized a two-layer MLP for the decoder layer and set $\lambda = 0.1$. Regarding the minimum coverage of the Relative Text-Position Prediction Task (RTPP), we set the minimum coverage $c_{\min}$ to 30% to ensure that the selected region is not too small.

We trained our model with a learning rate of $1 \times 10^{-4}$ for 10 epochs, incorporating a linear warmup of 50 steps, followed by cosine decay to 0. The total batch size was set to 256, and we conducted training on 8 A100 GPUs. The entire training time is approximately 2 days for the BLIP-2-based model and 5 days for the mPLUG-Owl-based model.

# E   Datasets

For all of the benchmarks, we used the train/val/split set provided by UReader [19], which is basically built on DUE benchmark [59].

**DocVQA**   DocVQA [45] consists of 50,000 question and answer (QA) pairs derived from 12,000 document images sourced from the UCSF Industry Documents Library. It comprises 40,000 training questions, 5,000 validation questions, and 5,000 test questions. The evaluation metric utilized is ANLS (Average Normalized Levenshtein Similarity), which measures similarity based on edit distance.

**InforgraphicsVQA**   The InfographicVQA dataset [46] comprises a total of 30,035 questions and 5,485 images sourced from 2,594 distinct web domains. The data is randomly split into 23,946 questions and 4,406 images for training, 2,801 questions and 500 images for validation, and 3,288 questions and 579 images for testing. These questions demand approaches that integrate reasoning across document layout, textual content, graphical elements, and data visualizations simultaneously. We use ANLS for the evaluation metric.

**DeepForm**   DeepForm [47] serves as an information extraction dataset comprising 1100 socially significant documents pertaining to election spending. The objective entails extracting contract numbers, advertiser names, payment amounts, and air dates from advertising disclosure forms submitted to the Federal Communications Commission. The benchmark is evaluated by F1 score.

**KleisterCharity**   KleisterCharity (KLC) [48] represents another information extraction dataset, featuring 2,700 documents sourced from published reports of charity organizations. It tackles significant challenges, including high variability in layout (due to the absence of templates), the necessity of performing Optical Character Recognition (OCR), the presence of lengthy documents, and the existence of various spatial features such as tables, lists, and titles. The evaluation metric utilized is F1 score.

**WikiTableQuestions**   WikiTableQuestions (WTQ) [49] comprises 2.1k table images from Wikipedia and is annotated with 23k question and answer pairs demanding comparison and arithmetic operations. The task necessitates deeper compositionality, leading to a combinatorial explosion within the logical forms space. We use accuracy for the evaluation metric.

**TabFact**   TabFact [50] is a Natural Language Inference dataset comprising 112k statements labeled as 'entailed' or 'refuted' based on information from 16k Wikipedia tables. The dataset is divided into 92,283 statements for training, 12,792 for validation, and 12,779 for testing. The evaluation metric utilized is accuracy.

**ChartQA**   ChartQA [51] is designed to assist users by taking a chart and a natural language question as input to predict the answer. The dataset comprises a total of 21,000 chart images and 32,000 QA pairs, divided into 28,200 for training, 1,920 for validation, and 2,500 for testing. We use relaxed accuracy for the evaluation metric.

---

[2] https://github.com/LukeForeverYoung/UReader.

Table 7: Results of the ablation studies on mPLUG-Owl-based model. Note that the reconstruction loss (Recon) only comes with MS + HVFA.

| MS+HVFA | Recon | RFT [19] | RPT | PTP | DocVQA | InfoVQA | DeepForm | KLC | WTQ | TabFact | ChartQA | VisualMRC | TextVQA | TextCaps |
|---|---|---|---|---|---|---|---|---|---|---|---|---|---|---|
| √ | | | | | 67.64 | 43.42 | 50.34 | 34.45 | 27.49 | 65.75 | 60.77 | 211.03 | 56.47 | 119.65 |
| √ | √ | | | | 68.14 | 43.93 | 50.44 | 34.58 | 29.02 | 66.19 | 59.25 | 214.91 | 56.76 | 120.55 |
| √ | | | √ | √ | 71.33 | 45.21 | 52.42 | **36.88** | 34.14 | 68.02 | 62.90 | 224.34 | 58.73 | 122.32 |
| √ | √ | √ | | √ | 71.77 | **45.92** | 51.89 | 36.37 | 33.71 | 67.86 | 62.35 | 225.60 | 59.16 | 122.79 |
| √ | √ | | √ | √ | **72.68** | 45.90 | **53.02** | 36.73 | **34.51** | 68.19 | 63.28 | 226.43 | **59.22** | **123.11** |

Table 8: Performance of the BLIP-2-based model with varying scales.

| # of scales | Throughput (img/s) | DocVQA | InfoVQA | DeepForm | KLC | WTQ | TabFact | ChartQA | VisualMRC | TextVQA | TextCaps |
|---|---|---|---|---|---|---|---|---|---|---|---|
| 1 | 1.481 | 40.18 | 25.19 | 6.49 | 18.97 | 17.80 | 56.85 | 38.88 | 219.74 | 47.54 | 128.39 |
| 2 | 1.088 | 51.38 | 29.60 | 14.58 | 23.78 | 21.15 | 59.87 | 50.41 | 228.65 | 57.32 | 135.24 |
| 3 | 0.494 | **52.78** | **31.25** | **20.59** | **25.42** | **23.21** | **60.12** | **52.21** | **230.22** | **60.22** | **137.57** |

**VisualMRC**  VisualMRC [52] consists of 5,000 full screenshots of webpages from 35 websites, with 30,000 annotated QA pairs. The answers, averaging 9.53 words, are expressed in fluent sentences and are notably longer than those in other QA datasets. The dataset emphasizes natural language understanding and generation, sourcing its questions and abstractive answers from a diverse range of webpage domains. We use CIDEr score for the evaluation metric.

**TextVQA**  TextVQA [53] dataset is designed for evaluating models' abilities to answer questions about images where the answers are textually present within the images. It consists of images from the OpenImagesV3 [60] dataset paired with questions and answers. The dataset comprises approximately 45,336 images, 28,408 training questions, 3,000 validation questions, and 7,684 test questions. We employ accuracy for the evaluation metric.

**TextCaps**  TextCaps [54] dataset is designed for the task of image captioning with a focus on reading and understanding text present in images. It consists of images sourced from the OpenImages dataset, annotated with captions that require the model to recognize and interpret text within the image. The dataset contains over 145,000 images and 230,000 captions. TextCaps challenges models to integrate visual and textual information to generate accurate and informative captions that include text found in the image. We employ the CIDEr score as our evaluation metric.

# F  Additional Ablation Study Results

To validate the effect of each component more robustly, we conduct the additional ablation study for the mPLUG-Owl-based model. We focused on components that might be considered most marginal: the reconstruction process and Relative Text-Position Prediction (RTPP). Note that we compare RPT to RFT for RTPP experiments. Table 7 shows that the reconstruction loss and RPTT consistently improve performance on most tasks with the large backbone model, mPLUG-Owl.

# G  Analysis on using more Scales

Our proposed method is inherently flexible, permitting extension to accommodate an arbitrary number of scales. For example, a hierarchical configuration utilizing three scales can be implemented by initially integrating features from scales 3 and 2, followed by merging these with features from scale 1. Table 8 presents the performance across various benchmarks for differing numbers of scales. This experiment employed the BLIP-2-based variant of our model. The findings reveal that the inclusion of a third scale continues to enhance document comprehension, albeit with diminishing returns. This is partly because, as the model has already captured most of the key information from the initial two scales, the additional benefit is reduced. Note that training the model with all three scales without our hierarchical visual feature aggregation module is not feasible on A100 GPUs due to memory constraints.

Table 9: Comparison with task-specific state-of-the-arts methods.

| | DocVQA | InfoVQA | DeepForm | KLC | WTQ |
|---|---|---|---|---|---|
| SoTA | 92.3 (InternVL-1.5-Plus) | 75.7 (InternVL-1.5-Plus) | 68.8 (mPLUG-DocOwl 1.5) | 38.7 (mPLUG-DocOwl 1.5) | 40.6 (mPLUG-DocOwl 1.5) |
| Ours | 72.7 | 45.9 | 53.0 | 36.7 | 34.5 |

| | TabFact | ChartQA | VisualMRC | TextVQA | TextCaps |
|---|---|---|---|---|---|
| SoTA | 80.4 (mPLUG-DocOwl 1.5) | 81.3 (ChartPaLI) | 364.2 (LayoutT5) | 82.2 (Omni-SMoLA) | 164.3 (PaLI-3) |
| Ours | 68.2 | 63.3 | 226.4 | 59.2 | 123.1 |

# H  Comparison with Task-specific SoTA Methods

We list the performance of task-specific state-of-the-art models at the moment of submission in Table 9. The state-of-the-art models can be categorized into 3 categories.

**Large Foundation Models**   These models leverage extensive pretraining datasets and large model parameters. For DocVQA and InfographicsVQA, InternVL-1.5-Plus [61] ranks highest on the leaderboard. This model has 26B parameters and is pretrained on 26 dataset corpora before being fine-tuned on an additional 49 corpora. For TextVQA and TextCaps, variants of PaLI [62, 63] hold the top rank. PaLI, which is trained on the 10B-WebLI dataset, is a significantly stronger foundation model than our backbone, mPLUG-Owl, which only uses a 1B dataset for pretaining. While PaLI-3 [63] has 5B parameters, it still benefits from the 10B-WebLI dataset for pretaining. Moreover, direct comparison becomes more challenging when considering the differences in the unimodal encoders of each model. Note that we have compared our model only with OCR-free methods based on comparable MLLMs in terms of model size and dataset scale.

**Methods Using OCR Engines**   For VisualMRC, current OCR-free models lag behind LayoutT5 [52], which utilizes external OCR engines. This is likely because VisualMRC data is text-rich and features long documents, indicating significant room for improvement. While LayoutT5 has a relatively small model size of 770M parameters compared to recent MLLMs, incorporating OCR engines in such task provides a substantial advantage, particularly in accurately processing and understanding extensive textual information.

**Fine-Tuning Foundation Models with Document-Specific Data and Methods**   Our work falls into this category. For DeepForm, KLC, WTQ, and TabFact, the current SoTA model is mPLUG-DocOwl 1.5 [64], which is a concurrent work to ours. mPLUG-DocOwl 1.5 is based on mPLUG-Owl 2 [65], a direct extension of our backbone, mPLUG-Owl. While the models are similar in size, mPLUG-Owl 2 is known to perform significantly better than mPLUG-Owl. Additionally, mPLUG-DocOwl 1.5 is fine-tuned on 4M document datasets, which is several times more than our 650K dataset. For ChartQA, where ChartPaLI [66] is the state-of-the-arts, instruction data specifically designed for chart data is used, and the model is based on PaLI, pretrained with the 10B-WebLI dataset.

# I  Instruction Templates for Relative Text-Position Prediction Task

Table 10 illustrates the full list of the templates for Relative Text-Position Prediction Task.

Table 10: Examples for Relative Text-Position Prediction Task

| Task | Instruction Templates |
|------|----------------------|
| RPT (first) | Human: What's in the first { }% of the image text? AI: {corresponding texts}.
Human: Identify words from the first { }% of the image text. AI: {corresponding texts}.
Human: Which words make up the first { }% of the text in the image? AI: {corresponding texts}.
Human: List words in the image text's initial { }% segment. AI: {corresponding texts}.
Human: Extract words found in the image text's opening { }%. AI: {corresponding texts}.
Human: Words in the initial { }% of the image text? AI: {corresponding texts}.
Human: Which words comprise the image text's first { }%? AI: {corresponding texts}.
Human: Identify the words positioned in the image text's initial { }%. AI: {corresponding texts}.
Human: List the words found within the first { }% of the image text. AI: {corresponding texts}.
Human: What words are situated within the initial { }% segment of the image text? AI: {corresponding texts}. |
| RPT (middle) | Human: What are the words located between { }% and { }% of the text in the image? AI: {corresponding texts}.
Human: List words found between { }% and { }% in the image text. AI: {corresponding texts}.
Human: Which words fall between { }% and { }% in the image text? AI: {corresponding texts}.
Human: What words are in the { }%-{ }% range of the image text? AI: {corresponding texts}.
Human: Identify words from { }%-{ }% in the image text. AI: {corresponding texts}.
Human: What are the words located at { }%-{ }% in the image text? AI: {corresponding texts}.
Human: Can you extract words from { }%-{ }% in the image text? AI: {corresponding texts}.
Human: What's within the { }%-{ }% range in the image text? AI: {corresponding texts}.
Human: Which words occupy { }%-{ }% of the image text? AI: {corresponding texts}.
Human: What words lie between { }% and { }% in the image text? AI: {corresponding texts}. |
| RPT (last) | Human: Identify words from the last { }% of the image text. AI: {corresponding texts}.
Human: Which words make up the last { }% of the text in the image? AI: {corresponding texts}.
Human: What's in the last { }% of the image text? AI: {corresponding texts}.
Human: List words in the image text's final { }% segment. AI: {corresponding texts}.
Human: Extract words found in the image text's closing { }%. AI: {corresponding texts}.
Human: Words in the final { }% of the image text? AI: {corresponding texts}.
Human: Which words comprise the image text's last { }%? AI: {corresponding texts}.
Human: Identify the words positioned in the image text's final { }%. AI: {corresponding texts}.
Human: List the words found within the last { }% of the image text. AI: {corresponding texts}.
Human: What words are situated within the final { }% segment of the image text? AI: {corresponding texts}. |
| PTP | Human: Specify the relative position within the image where {query texts} is found. AI: { }%-{ }%.
Human: Where is the text {query texts} located within the image? AI: { }%-{ }%.
Human: Locate the relative position within the image where the text {query texts} is situated. AI: { }%-{ }%.
Human: Determine the relative position within the image where the words {query texts} appear. AI: { }%-{ }%.
Human: Identify the relative position within the image where the phrase {query texts} is located. AI: { }%-{ }%.
Human: Find the relative position within the image where {query texts} is depicted. AI: { }%-{ }%.
Human: Where within the image can we find the phrase {query texts}? AI: { }%-{ }%.
Human: At what position in the image do the words {query texts} appear? AI: { }%-{ }%.
Human: Where within the image is {query texts} depicted? AI: { }%-{ }%.
Human: Where can we locate {query texts} within the image? AI: { }%-{ }%. |

