# OpenReview forum: "Hierarchical Visual Feature Aggregation for OCR-Free Document Understanding"
_NeurIPS.cc/2024/Conference — NeurIPS 2024 poster_

### Official Review · Reviewer_Pfd9 · 2024-07-11

**Soundness:** 3
**Presentation:** 3
**Contribution:** 2
**Rating:** 4
**Confidence:** 5

**Summary:**

This paper introduces an approach to enhancing document understanding capabilities by employing a hierarchical visual feature aggregation technique alongside pretrained Multimodal Large Language Models (MLLMs). The method utilizes a feature pyramid hierarchy integrated with cross-attentive pooling, which effectively captures multi-scale visual information. Additionally, the paper presents a new instruction task designed to further boost model performance. The experimental results on standard benchmarks demonstrate the effectiveness and superiority of the proposed method in advancing document understanding.

**Strengths:**

1. This paper conducts extensive ablation studies to meticulously demonstrate the effectiveness of each proposed module and parameter. These studies provide a comprehensive understanding of the individual contributions and interactions within the framework, thereby validating the design choices and overall robustness of the system.

2. This paper introduces a novel relative text-position prediction task, which significantly enhances document understanding. This task leverages the spatial relationships between text elements to improve the model's ability to interpret and contextualize information within documents.

**Weaknesses:**

1. The baseline for the ablation studies presented in this paper may be considered less robust, as evidenced by the performance on DocVQA, where it only achieves 35.17% accuracy, significantly below the 72.7% achieved by the proposed model. This discrepancy suggests that the baseline's performance could potentially undermine the persuasiveness of the conclusions drawn from the ablation study. It is possible that certain modules or techniques, which show effectiveness against this baseline, may not demonstrate the same level of impact when compared to a stronger baseline.

2. While the multi-scale feature pyramid hierarchy is identified as the paper's primary contribution, the discussion surrounding this module could be more thorough. For instance, the paper could benefit from an exploration of alternative configurations, such as the implications of utilizing only two local scales versus three or four. A deeper analysis of these options would provide insights into the optimal balance between computational efficiency and performance, thereby enriching the understanding of the module's role in the overall framework.

**Questions:**

Please see the concerns in Weaknesses.

Other questions:

Whether the "Ours" in Table 3 applied the relative text-position prediction task?

**Limitations:**

No.

---

> ### Author Rebuttal · Authors · 2024-08-07
>
> We are grateful for the reviewer's insightful recognition of our extensive ablation studies and the novel relative text-position prediction task, which validate our approach's robustness. Below are our responses to the main concerns.
>
> **[W1] Robustness of the ablation studies**
>
> As 35.17% represents the performance of the moderate BLIP2-OPT backbone and 72.7% represents the performance of the larger mPLUG-Owl backbone on DocVQA, they are not directly comparable. To provide the proper baseline, we brought the numbers for the mPLUG-Owl baseline in Table D (row 5) from the ablation study of UReader paper [14]. To further address the concern about the robustness of our ablation studies, we have conducted additional experiments on the mPLUG-Owl-based model. Due to the limited time constraints in the rebuttal period, we focused on components that might be considered most marginal: the reconstruction process (row 2 vs row 4, row 6 vs row 8) and Relative Text-Position Prediction (RTPP) (row 3 vs row 4, row 7 vs row 8). Note that we compare RPT to RFT for RTPP experiments. As shown in Table D, the reconstruction loss and RPTT consistently improve performance on most tasks with both the moderate and large backbone models, BLIP2-OPT and mPLUG-Owl, respectively. We believe these consistent gains demonstrate the effectiveness of our components. We will provide the full ablation studies for the large model in the revised version.
>
> Table D: Results of ablation studies on the reconstruction loss and RTPP. Note that we compare RPT to RFT for RTPP experiments. The last row for each backbone model represents our final model.
>
> | Backbone  | MS+HVFA | Recon | RFT | RPT | PTP |   DocVQA  |  InfoVQA  |  DeepForm |    KLC    |    WTQ    |  TabFact  |  ChartQA  |  VisualMRC |  TextVQA  |  TextCaps  |
> |-----------|:-------:|:-----:|:---:|:---:|:---:|:---------:|:---------:|:---------:|:---------:|:---------:|:---------:|:---------:|:----------:|:---------:|:----------:|
> | BLIP2-OPT |         |       |     |     |     |   35.17   |   20.34   |    4.14   |   18.50   |   15.36   |   53.37   |   31.76   |   201.39   |   43.20   |   126.34   |
> | BLIP2-OPT |    ✓    |       |     |  ✓  |  ✓  |   50.46   |   28.48   |   13.18   | **23.82** |   20.06   |   59.83   |   49.48   |   226.65   |   56.04   |   134.06   |
> | BLIP2-OPT |    ✓    |   ✓   |  ✓  |     |  ✓  |   50.62   |   29.49   |   12.55   |   22.88   |   20.51   |   59.28   |   48.41   |   226.55   |   57.19   |   134.44   |
> | BLIP2-OPT |    ✓    |   ✓   |     |  ✓  |  ✓  | **51.38** | **29.60** | **14.58** |   23.78   | **21.15** | **59.87** | **50.41** | **228.65** | **57.32** | **135.24** |
> | mPLUG-Owl |         |       |     |     |     |    56.7   |    ---    |    ---    |    ---    |    22.9   |    ---    |    56.7   |    205.0   |    54.3   |     ---    |
> | mPLUG-Owl |    ✓    |       |     |  ✓  |  ✓  |   71.33   |   45.21   |   52.42   | **36.88** |   34.14   |   68.02   |   62.90   |   224.34   |   58.73   |   122.32   |
> | mPLUG-Owl |    ✓    |   ✓   |  ✓  |     |  ✓  |   71.77   | **45.92** |   51.89   |   36.37   |   33.71   |   67.86   |   62.35   |   225.60   |   59.16   |   122.79   |
> | mPLUG-Owl |    ✓    |   ✓   |     |  ✓  |  ✓  | **72.68** |   45.90   | **53.02** |   36.73   | **34.51** | **68.19** | **63.28** | **226.43** | **59.22** | **123.11** |
>
> **[W2] Analysis on using more scales**
>
> We appreciate the reviewer's suggestion to explore alternative configurations of our multi-scale feature pyramid hierarchy. Our method is designed to be flexible and can be extended to use an arbitrary number of scales. For example, a 3-scale configuration with feature hierarchy can be implemented as follows: combining features from scales 3 and 2, then merging the results with features from scale 1. Due to significant time constraints during the rebuttal period, it is challenging to conduct extensive experiments on these alternatives. We are currently running an experiment with the model that integrates 3 scales. We present the validation performance of each task after 2 epochs in Table E.
>
> Table E. Validation performance with varying scales after 2 epochs.
>
> | # of scales | Throughput (img/s) |   DocVQA  |  InfoVQA  |  DeepForm |    KLC    |    WTQ    |  TabFact  |  ChartQA  |  VisualMRC |  TextVQA  |  TextCaps  |
> |:-----------:|:------------------:|:---------:|:---------:|:---------:|:---------:|:---------:|:---------:|:---------:|:----------:|:---------:|:----------:|
> |      1      |        1.481       |   36.29   |   21.24   |   16.62   |   36.23   |   14.31   |   52.52   |   27.44   |   178.04   |   42.63   |   126.56   |
> |      2      |        1.088       |   42.18   |   24.53   |   21.26   |   38.74   |   16.60   |   53.13   |   32.81   |   195.55   |   45.22   |   129.24   |
> |      3      |        0.494       | **44.86** | **27.57** | **25.61** | **39.48** | **18.77** | **54.17** | **33.05** | **201.49** | **46.92** | **131.82** |
>
> When integrating the 3rd scale in addition to the 1st and 2nd scale, the number of visual inputs increases about $(1+g+4g+16g)/(1+g+4g) ≈ 4.2$ times, while the throughput of our model only decreases by a factor of 2. Note that $g$ represents the number of grids in line 124. The validation results demonstrate that incorporating the 3rd scale still helps the document understanding but the improvement gain is diminished. This is expected because, as more scales are added, the incremental benefit decreases since the model has already captured most of the relevant information from the initial scales. We expect this experiment to be finished within the discussion period and will post the performance on the test set.
>
> **[Q1] Clarification on Table 3**
>
> Yes. The results labeled as "Ours" in Table 3 indeed represent our final model, which includes the Relative Text-Position Prediction Task.

---

> > ### Comment · Reviewer_Pfd9 · 2024-08-12
> > **Final rating**
> >
> > Thanks for the rebuttal. After reading the other reviews and the rebuttal, I would like to keep my rating. Although the rebuttal addressed some of my concerns, such as the analysis and the ablation studies, I agree with Reviewer CDZQ that the multi-scale features integration is not novel.

---

> ### Author Response · Authors · 2024-08-12
> **Final test results for using more scales**
>
> Table F. Test scores of the BLIP-2-based model with varying scales.
>
> | # of scales | Throughput (img/s) |   DocVQA  |  InfoVQA  |  DeepForm |    KLC    |    WTQ    |  TabFact  |  ChartQA  |  VisualMRC |  TextVQA  |  TextCaps  |
> |:-----------:|:------------------:|:---------:|:---------:|:---------:|:---------:|:---------:|:---------:|:---------:|:----------:|:---------:|:----------:|
> |      1      |        1.481       |   40.18   |   25.19   |    6.49   |   18.97   |   17.80   |   56.85   |   38.88   |   219.74   |   47.54   |   128.39   |
> |      2      |        1.088       |   51.38   |   29.60   |   14.58   |   23.78   |   21.15   |   59.87   |   50.41   |   228.65   |   57.32   |   135.24   |
> |      3      |        0.494       | **52.78** | **31.25** | **20.59** | **25.42** | **23.21** | **60.12** | **52.21** | **230.22** | **60.22** | **137.57** |
>
> The experiments with the 3-scale variant of our model have been completed and the test scores for the different number of scales across each benchmark are presented in Table F. We used the BLIP-2-based version of our model and all models are trained using RTPP for the text reading task for this experiment. The results show that incorporating an additional 3rd scale continues to help with document understanding, but the improvement gain becomes less significant. This is understandable, as the model has already captured most of the key information from the initial two scales, so the additional benefit is reduced. Note that training the model with all three scales without our hierarchical feature aggregation module is not feasible on A100 GPUs due to memory constraints. We will revise our manuscript to include this analysis.

---

> ### Author Response · Authors · 2024-08-12
>
> Dear Reviewer Pfd9
>
> Because the end of discussion period is approaching, we kindly ask you whether our response is helpful to clarify you or not. Also, if you have any question or additional comments, please do not hesitate to contact us. We thank you for your time and efforts to review our paper.
>
> Best wishes,
>
> Authors.

---

> ### Author Response · Authors · 2024-08-12
> **Clarification on the contribution of our work**
>
> Thanks for your timely response. As we stated in our rebuttal to Reviewer CDZQ, our main contribution lies not only in using multi-scale features, but also in **how we aggregate them without significantly increasing computational costs**. This is an important issue for MLLMs. While multi-scale features clearly benefit MLLMs, it is not computationally feasible without an appropriate strategy. By addressing this challenge, our approach enables broader applicability of MLLMs across different hardware environments.

---

> > ### Author Response · Authors · 2024-08-13
> >
> > Dear Reviewer Pfd9,
> >
> > We sincerely appreciate your time and effort in reviewing our work. Given that Reviewer CDZQ has acknowledged the value of the hierarchical feature aggregation module as a meaningful contribution to multi-scale feature aggregation, we kindly ask if you could reconsider the contribution of our work in this light.
> >
> > Thank you for your consideration.

---

### Official Review · Reviewer_rxzX · 2024-07-12

**Soundness:** 3
**Presentation:** 3
**Contribution:** 2
**Rating:** 5
**Confidence:** 4

**Summary:**

This paper presents hierarchical visual feature aggregation for OCR-free document understanding, leveraging feature pyramid hierarchy with cross-attentive pooling to handle the trade-off between information loss and efficiency. Additionally, a relative text-position prediction task is proposed to address the text truncation issue. Experiment results based on BLIP-2 and mPLUG-Owl demonstrate the effectiveness of the proposed method.

**Strengths:**

1. This paper innovatively utilizes a feature pyramid hierarchy to fuse multi-scale visual features without increasing the computational complexity of large language models.
2. This paper presents a novel instruction tuning task that is robust to text truncation issues.
3. Extensive experiments and ablation studies showcase the effectiveness of the proposed method on BLIP-2 and mPLUG-Owl models.

**Weaknesses:**

1. The comparison in Table 3 is not comprehensive enough; models such as mPLUG-DocOwl 1.5 and TextMonkey should be included.
2. In the ablation experiments based on BLIP-2, the performance improvements brought by the reconstruction layer (Table 4) and the Relative Text-Position Prediction Task (Table 6, compared to Reading Full Text) are minimal. I wonder whether they can work for more advanced LMMs such as mPLUG-Owl.

**Questions:**

1. In line 131, the description "augmenting each grid by a 2 × 2 scale, effectively enlarging the receptive field size" is confusing. How does this operation lead to an enlarged receptive field size?
2. How are the transposition and matrix multiplication operations performed in Equation 2 given the 4-dimensional matrix F_{i+1} and F^’_{i+1}.

**Limitations:**

The authors did not validate their proposed methods on more advanced models such as mPLUG-DocOwl 1.5 and TextMonkey, so the generalizability of their methods remains open to discussion.

---

> ### Author Rebuttal · Authors · 2024-08-07
>
> We truly thank the reviewer for recognizing our key innovations in multi-scale feature fusion, novel instruction tuning, and comprehensive experimental results across different models. Below are our responses to the main concerns.
>
> **[W1] Comparison to the recent works**
>
> Table B: Comparison of different configurations over TextMonkey [R4] and mPLUG-Owl 1.5 [R5].
>
>
> | Method           | Backbone    | # Model Params | # Trainable Params | # Pretraining Data | # Fine-tuning data |
> |------------------|-------------|----------------|--------------------|--------------------|--------------------|
> | TextMonkey [R4]      | Qwen-VL [25]    | 9.7B           | 7.9B               | 1.4B+76.8M         | 2.1M               |
> | mPLUG-DocOwl 1.5 [R5] | mPLUG-Owl 2 [R6] | 8.1B           | 412M               | 400M               | 4M+25K             |
> | Ours             | mPLUG-Owl   | 7.2B           | 96M                | 1.1B               | 650K               |
>
> Initially, we compare the configurations with our framework and two other approaches, TextMonkey and mPLUG-Owl 1.5, represented in Table B. We would like to highlight two key differences between both models and ours. Both models use more advanced Multimodal Large Language Model (MLLM) backbones for initialization - Qwen-VL and mPLUG-Owl2. Our model, in contrast, is based on the mPLUG-Owl. Additionally, both models benefit from substantially larger training datasets and more trainable parameters, which directly impacts their performance.
>
> Despite these advantages, according to the reported results, our method surpasses TextMonkey. Although the mPLUG-DocOwl 1.5 model performs slightly better than ours, this is possible given its use of a more advanced backbone and larger dataset. For a fair comparison, the best option would be to augment both backbones with our framework and train the models with the same data, which is not feasible within a limited rebuttal period. However, we expect improvements in our results by incorporating our framework into the new backbones with more data, since the multi-scale aggregation module facilitates efficient capturing of different levels of detail.
>
> **[W2] Effect of the reconstruction loss and RTPP on mPLUG-Owl**
>
> Table C: Results of ablation studies on the reconstruction loss and RTPP. Note that we compare RPT to RFT for RTPP experiments. The last rows for each backbone model represent our final model.
>
> | Backbone  | MS+HVFA | Recon | RFT | RPT | PTP |   DocVQA  |  InfoVQA  |  DeepForm |    KLC    |    WTQ    |  TabFact  |  ChartQA  |  VisualMRC |  TextVQA  |  TextCaps  |
> |-----------|:-------:|:-----:|:---:|:---:|:---:|:---------:|:---------:|:---------:|:---------:|:---------:|:---------:|:---------:|:----------:|:---------:|:----------:|
> | BLIP2-OPT |    ✓    |       |     |  ✓  |  ✓  |   50.46   |   28.48   |   13.18   | **23.82** |   20.06   |   59.83   |   49.48   |   226.65   |   56.04   |   134.06   |
> | BLIP2-OPT |    ✓    |   ✓   |  ✓  |     |  ✓  |   50.62   |   29.49   |   12.55   |   22.88   |   20.51   |   59.28   |   48.41   |   226.55   |   57.19   |   134.44   |
> | BLIP2-OPT |    ✓    |   ✓   |     |  ✓  |  ✓  | **51.38** | **29.60** | **14.58** |   23.78   | **21.15** | **59.87** | **50.41** | **228.65** | **57.32** | **135.24** |
> | mPLUG-Owl |    ✓    |       |     |  ✓  |  ✓  |   71.33   |   45.21   |   52.42   | **36.88** |   34.14   |   68.02   |   62.90   |   224.34   |   58.73   |   122.32   |
> | mPLUG-Owl |    ✓    |   ✓   |  ✓  |     |  ✓  |   71.77   | **45.92** |   51.89   |   36.37   |   33.71   |   67.86   |   62.35   |   225.60   |   59.16   |   122.79   |
> | mPLUG-Owl |    ✓    |   ✓   |     |  ✓  |  ✓  | **72.68** |   45.90   | **53.02** |   36.73   | **34.51** | **68.19** | **63.28** | **226.43** | **59.22** | **123.11** |
>
> In response to concerns about the robustness of reconstruction loss and Relative Text-Position Prediction (RTPP) on the advanced backbone model, we conducted additional ablation studies on the mPLUG-Owl-based model, which are presented in Table C. Note that we compare RPT to RFT for RTPP experiments. The results demonstrate that both reconstruction loss and RTPP consistently improve performance on most tasks with both the moderate and large backbone models, BLIP2-OPT and mPLUG-Owl, respectively. We believe these consistent gains demonstrate the effectiveness of our components.
>
> **[Q1] Clarification on the receptive field**
>
> In line 131, the description “effectively enlarging the receptive field size” should have been removed. We are sorry for the confusion. To clarify, augmenting each grid by a 2 × 2 scale does not enlarge the receptive field as originally stated. Instead, this operation provides a more detailed view of each grid area by effectively zooming in, allowing our model to capture finer-grained visual features and smaller text fonts within each cell. This increased resolution of information potentially leads to a better understanding of local details. We will revise this description to accurately reflect the purpose and effect of this operation.
>
> **[Q2] Clarification of Matrix multiplication**
>
> We simply flatten both matrices to the shape of $(H_i \times W_i \times Q) \times C$. We will revise the description to provide a clearer explanation.
>
> [R4] Liu et al., TextMonkey: An OCR-Free Large Multimodal Model for Understanding Document, arXiv 2024
>
> [R5] Hu et al., mPLUG-DocOwl 1.5: Unified Structure Learning for OCR-free Document Understanding, arXiv 2024
>
> [R6] Ye et al., mPLUG-Owl2: Revolutionizing Multi-modal Large Language Model with Modality Collaboration, CVPR 2024

---

> > ### Comment · Reviewer_rxzX · 2024-08-11
> >
> > Thank you for your repsonses. I would like to maintain the original rating.

---

> > > ### Author Response · Authors · 2024-08-12
> > >
> > > Dear Reviewer rxzX,
> > >
> > > We thank you for giving our paper a positive score. We will revise the main paper to reflect your comments and discussions. If you have any question or additional comments, please do not hesitate to contact us.
> > >
> > > Best wishes,
> > >
> > > Authors.

---

### Official Review · Reviewer_CDZQ · 2024-07-12

**Soundness:** 3
**Presentation:** 2
**Contribution:** 2
**Rating:** 5
**Confidence:** 5

**Summary:**

The paper analyzes the impact of features at different scales for document understanding. It proposes a method to combine multi-scale features without signifficantly increasing computational complexity. In addition the paper also proposes two new instruction tuning tasks that allow the model to better extract text information from the documents. Experiments are reported on a varied set of datasets and tasks showing

**Strengths:**

- The paper introduces multi-scale feature extraction for document understanding. This can help to analyze larger and high resolution documents, without signifficantly increasing the cost.
- Two new instruction tuning tasks are defined to help the model learn how to read the text inside the document.
- Experimental results show that combining the proposed approach with existing MLLMs obtain better results than other existing methods. A detailed ablation study shows the contribution of all the components of the framework.

**Weaknesses:**

- From a technical point of view there is not much novelty. Multi-scale feature integration has been largely been used in many different domains in a similar way.
- Experimental results lack a better comparison with SoA models. The proposed method is compared with OCR-free models and similar approaches using MLLMs, but ignores other methods of the SoA (maybe using an external OCR) that can obtain better results. For instance, in the leadeboard of DocVQA and InfographicVQA (rrc.cvc.uab.es) there are methods with a better performance. For a fair analysis of the results, I think that these results should be included and discussed comparing and contextualizing with the results obtained by the proposed method. Also, for tasks that mainly work with natural images, such as TextVQA, it would be better to compare with specific methods for scene text VQA. Comparing with methods designed to work with document images like Donut may not be the best option.

**Questions:**

See above in weaknesses

**Limitations:**

There is no specific discussion on the limitations of the method.

---

> ### Author Rebuttal · Authors · 2024-08-07
>
> We are deeply grateful for the reviewer's insightful summary of our work, highlighting the key contributions of our multi-scale feature extraction approach, new instruction tuning tasks, and comprehensive experimental results. Below are our responses to the main concerns.
>
> **[W1] Incremental novelty (multi-scale features)**
>
> The proposed approach is the first practical solution for integrating multi-scale features in high-resolution multi-modal large language models (MLLMs), which was deemed impractical due to resource constraints. Our framework preserves the benefits of multi-scale features without substantial increases in computational costs. Specifically, we developed a hierarchical visual feature aggregation module that substantially reduces computational costs using cross-attentive pooling and feature pyramid hierarchy, making multi-scale integration computationally feasible for MLLMs. Unlike prior works relying on single-scale inputs, our method allows efficient multi-scale processing in MLLMs, significantly improving performance.
>
> **[W2] Justification on comparisons**
>
> We appreciate the reviewer's comments but respectfully and partly disagree that our comparisons are lacking, due to the following reasons:
>
>  Our research focuses on OCR-free multi-task learning with MLLMs for recognizing document images (e.g., DocVQA) and understanding natural images (e.g., TextVQA), which is different from task-specific learning and OCR-based learning. Direct comparisons with task-specific models would not fully capture this key aspect of our work. For instance, ChartPaLI [R1], the state-of-the-art for ChartQA, uses instruction data specifically designed for chart data; it doesn't align with our general-purpose approach.
>
> On the other hand, the top performers on the referenced leaderboards, such as those for DocVQA and InfographicVQA, utilize significantly larger models, advanced language models with 20-55B parameters such as InternLM2 [R2] and PaLI-X [R3]. These are not directly comparable to our LLaMA-based mPLUG-Owl model due to significant differences in scale and computational costs. Moreover, the size and origin of the massive training data used for these large language models are often not fully disclosed, making direct comparisons potentially unfair. Hence, we compared our model only with OCR-free methods based on comparable MLLMs in terms of model size and dataset scale.
>
> We included Donut as a baseline, a well-established OCR-free approach, to demonstrate the effectiveness of our approach; Donut has an advantage in document-related tasks because it is pretrained with document-specific data, but our method is still competitive without such a specialized pretaining technique.
>
> [R1] Carbune et al., Chart-based Reasoning: Transferring Capabilities from LLMs to VLMs, arXiv 2024
>
> [R2] Cai et al., InternLM2 Technical Report, arXiv 2024
>
> [R3] Chen et al., PaLI-X: On Scaling up a Multilingual Vision and Language Model, arXiv 2023

---

> ### Author Response · Authors · 2024-08-12
>
> Dear Reviewer CDZQ
>
> Because the end of discussion period is approaching, we kindly ask you whether our response is helpful to clarify you or not. Also, if you have any question or additional comments, please do not hesitate to contact us. We thank you for your time and efforts to review our paper.
>
> Best wishes,
>
> Authors.

---

> > ### Comment · Reviewer_CDZQ · 2024-08-13
> >
> > I thank the authors for your responses to my comments. I agree that the hierarchichal feature aggregation module is a useful contribution for multi-scale feature aggregation. Concerning the comparison with the SoA I agree that there are different types of methods that are not fully comparable in terms of parameters or training data, but I think that when comparing with SoA all types of methods should be included with a specific discussion on the advantages of the proposed method even if it does not get the best results. The proposed method cannot be the best performing in terms of accuracy but it can have other positive aspects (ocr-free, number of parameters, data required to train) than can be remarked in the discussion.
> > Overall, I can raise a bit my original rating.

---

> ### Author Response · Authors · 2024-08-13
>
> Thank you for your thoughtful feedback and for recognizing the contribution of our hierarchical feature aggregation module. We appreciate your suggestion regarding the comparison with state of the art (SoTA) methods.
>
> Table G: Comparison with task-specific SoTA methods.
>
> |      |           DocVQA          |          InfoVQA          |         DeepForm        |           KLC           |           WTQ           |         TabFact         |      ChartQA     |     VisualMRC    |       TextVQA      |    TextCaps    |
> |------|:-------------------------:|:-------------------------:|:-----------------------:|:-----------------------:|:-----------------------:|:-----------------------:|:----------------:|:----------------:|:------------------:|:--------------:|
> | SoTA | 92.34 (InternVL-1.5-Plus) | 75.74 (InternVL-1.5-Plus) | 68.8 (mPLUG-DocOwl 1.5) | 38.7 (mPLUG-DocOwl 1.5) | 40.6 (mPLUG-DocOwl 1.5) | 80.4 (mPLUG-DocOwl 1.5) | 81.3 (ChartPaLI) | 364.2 (LayoutT5) | 82.22 (Omni-SMoLA) | 164.3 (PaLI-3) |
> | Ours |                      72.7 |                      45.9 |                      53.0 |                    36.7 |                    34.5 |                    68.2 |             63.3 |            226.4 |               59.2 |          123.1 |
>
> We list the performance of state-of-the-art models at the moment of submission in Table G. The SoTA models can be categorized into 3 categories.
>
> **Large Foundation Models**: These models leverage extensive pretraining datasets and large model parameters. For DocVQA and InfographicsVQA, InternVL-1.5-Plus [R7] ranks highest on the leaderboard. This model has 26B parameters and is pretrained on 26 dataset corpora before being fine-tuned on an additional 49 corpora. However, the exact amount of training data is not specified in their technical report. Additionally, QwenVL-Max is currently the SoTA for DocVQA, though technical details for this model are unavailable. For TextVQA and TextCaps, variants of PaLI [R8, R9] hold the top rank. PaLI, which is trained on the 10B-WebLI dataset, is a significantly stronger foundation model than our backbone, mPLUG-Owl, which only uses a 1B dataset for pretraining. While PaLI-3 [R9] has 5B parameters, it still benefits from the 10B-WebLI dataset for pretraining. Moreover, direct comparison becomes more challenging when considering the differences in the unimodal encoders of each model. InternVL-1.5-Plus consists of InternViT-6B and InternLM2-20B, while PaLI uses ViT-G/14 and UL-2 for vision and language processing, respectively. In contrast, our backbone, mPLUG-Owl, utilizes ViT-L/14 and LLaMA, which are considerably weaker than InternVL-1.5 and PaLI variants. Once again, we have compared our model only with OCR-free methods based on comparable MLLMs in terms of model size and dataset scale.
>
> **Methods Using OCR Engines**: For VisualMRC, current OCR-free models lag behind LayoutT5 [R10], which utilizes external OCR engines. This is likely because VisualMRC data is text-rich and features long documents, indicating significant room for improvement. While LayoutT5 has a relatively small model size of 770M parameters compared to recent MLLMs, incorporating OCR engines in such tasks provides a substantial advantage, particularly in accurately processing and understanding extensive textual information.
>
> **Fine-Tuning Foundation Models with Document-Specific Data and Methods**: Our work falls into this category. For DeepForm, KLC, WTQ, and TabFact, the current SoTA model is mPLUG-DocOwl 1.5 [R5]. As mentioned in our response to reviewer rxzX, mPLUG-DocOwl 1.5 is based on mPLUG-Owl 2, a direct extension of our backbone, mPLUG-Owl. While the models are similar in size, mPLUG-Owl 2 is known to perform significantly better than mPLUG-Owl. Additionally, mPLUG-DocOwl 1.5 is fine-tuned on 4M document datasets, which is several times more than our 650K dataset. It is also worth noting that mPLUG-DocOwl 1.5 was uploaded to arXiv in March and can be considered concurrent with our work. For ChartQA, where ChartPaLI is the SoTA [R1] as mentioned in the initial rebuttal, instruction data specifically designed for chart data is used, and the model is based on PaLI, pretrained with the 10B-WebLI dataset.
>
> We hope that this response clarify you. If you have any question or additional comments, please do not hesitate to contact us.
>
> [R1] Carbune et al., Chart-based Reasoning: Transferring Capabilities from LLMs to VLMs, arXiv 2024
>
> [R5] Hu et al., mPLUG-DocOwl 1.5: Unified Structure Learning for OCR-free Document Understanding, arXiv 2024
>
> [R7] Chen et al., How Far Are We to GPT-4V? Closing the Gap to Commercial Multimodal Models with Open-Source Suites, arXiv 2024
>
> [R8] Wu et al., Omni-SMoLA: Boosting Generalist Multimodal Models with Soft Mixture of Low-rank Experts, arXiv 2024
>
> [R9] Chen et al., Pali-3 vision language models: Smaller, faster, stronger, arXiv 2023
>
> [R10] Tanaka et al., VisualMRC: Machine reading comprehension on document images, AAAI 2021

---

### Official Review · Reviewer_cWFY · 2024-07-15

**Soundness:** 3
**Presentation:** 4
**Contribution:** 2
**Rating:** 6
**Confidence:** 4

**Summary:**

The paper presents a novel approach to OCR-free document understanding using pre-trained Multimodal Large Language Models (MLLMs). The approach uses multi-scale visual features to handle different font sizes within document images. To address the high computational cost associated with multi-scale visual inputs for pre-trained MLLMs, the authors propose a hierarchical visual feature aggregation module. This module reduces the number of input tokens to LLMs by employing a feature pyramid hierarchy with cross-attentive pooling, thereby balancing information loss and efficiency without being affected by varying document image sizes. In addition, the paper introduces an innovative instruction tuning task that incorporates text position information within images, improving model readability and robustness to text truncation. Extensive experiments demonstrate the effectiveness of the framework in achieving superior document understanding performance in various tasks.

**Strengths:**

- The paper provides a detailed description of model sizes, trainable parameters and pre-training data size. This information is critical for real-world application of these models where size and efficiency are critical factors.
- The authors carry out a very detailed ablation study. For some aspects of the study, they ensured that the number of parameters remained constant, minimising bias and allowing a clearer understanding of the impact of each component.
- The paper introduces a novel hierarchical visual feature aggregation module and a new instruction tuning task.

**Weaknesses:**

- Complexity vs. gain: The use of reconstruction loss increases complexity for what appears to be a small performance gain, as shown in Table 5 (lambda=0).
- RTP Usage: While the Random Text Positional (RTP) approach is interesting for handling complex documents, it is only used during training. There is potential to optimise and use RTP during inference to further improve performance. See the questions.
- Lack of open source availability: The models and code are not open source.

**Questions:**

- RTP optimisation: The RTP method reads a random percentage of text using a common reading order. Would it be more efficient to read specific, complete sections of the document based on its structure, such as reading entire articles in a newspaper?
- Text position encoding: Can you provide more details on how text position is encoded in the Positional Text Positional (PTP) method?
- Table 3 Formatting: The best result is not highlighted in bold for the 1st and 3rd columns of Table 3.

Typos and errors:
- Line 169: "stop-gradient" should be corrected to "stop-gradient".
- Line 189: "Table 7" should probably be "Table 1".
- Line 243: RTPP is introduced but never explained.

**Limitations:**

Yes.

---

> ### Author Rebuttal · Authors · 2024-08-06
>
> We sincerely appreciate the reviewer's acknowledgment of our introduction of novel components, thorough ablation studies, and detailed model information. Below are our responses to the main concerns.
>
> **[W1] Effectiveness of reconstruction loss**
>
> Table A. Effectiveness of reconstruction loss on both models. The bold-faced numbers indicate the best performance in each column for each backbone model.
>
> |  Backbone | Recon |   DocVQA  |  InfoVQA  |  DeepForm |    KLC    |    WTQ    |  TabFact  |  ChartQA  |  VisualMRC |  TextVQA  |  TextCaps  |
> |:---------:|:-----:|:---------:|:---------:|:---------:|:---------:|:---------:|:---------:|:---------:|:----------:|:---------:|:----------:|
> | BLIP2-OPT |       |   50.46   |   28.48   |   13.18   | **23.82** |   20.06   |   59.83   |   49.48   |   226.65   |   56.04   |   134.06   |
> | BLIP2-OPT |   V   | **51.38** | **29.60** | **14.58** |   23.78   | **21.15** | **59.87** | **50.41** | **228.65** | **57.32** | **135.24** |
> | mPLUG-Owl |       |   71.33   |   45.21   |   52.42   | **36.88** |   34.14   |   68.02   |   62.90   |   224.34   |   58.73   |   122.32   |
> | mPLUG-Owl |   V   | **72.68** | **45.90** | **53.02** |   36.73   | **34.51** | **68.19** | **63.28** | **226.43** | **59.22** | **123.11** |
>
> The reconstruction is performed only during training, so it incurs no additional cost for inference. Although the introduction of reconstruction loss increases training complexity, the extra cost is marginal, particularly compared to other components in our training strategies. More importantly, as shown in Table A from our ablation study, the reconstruction loss consistently improves performance on most tasks with both the moderate and large backbone models, BLIP2-OPT and mPLUG-Owl, respectively. We believe these consistent gains demonstrate the effectiveness of our reconstruction loss, especially given that the inference cost remains unchanged.
>
>
> **[W2 & Q1] Using Relative Text-Position Prediction (RTPP)**
>
> This comment including the associated question is a bit confusing partly due to the wrong terminology. It would be appreciated if you revise the comments for better answers. Doing the best under the current context, we presume that "Random Text Positional (RTP)" means Relative Text-Position Prediction (RTPP) in our approach. Using RTPP during inference is an interesting idea, but it may not be straightforward because it may require additional modules for generating proper instruction tasks or increase inference costs significantly; it can be a direction for the extension of our paper.
>
> **[W3] Lack of open source availability**
>
> We understand the concern about open-source availability. Due to the proprietary issue, we have to go through bureaucratic procedures to release the full source code. However, we will put our every effort into scientific transparency and reproducibility. We have already provided detailed implementation details in Section D in the Appendix. We will further provide extra details and ensure that our work is independently verified by the research community.
>
> **[Q2] More detail on Predicting Text Positional (PTP) task**
>
> The PTP task follows the same process as the Reading Partial Text (RTP) task for generating the position pair ($p_{start}$, $p_{end}$) and its corresponding text segment. The key difference lies in how to use this information:
> 1) RTP: The position pair is a part of the instruction, and the model predicts the text.
> 2) PTP: This is opposite to RTP: the text segment is given as the instruction, and the model predicts the position pair.
>
> In both cases, the position is represented by ($p_{start}$, $p_{end}$), indicating the start and end positions of the text segment within the document. We hope this clarifies the encoding process. We are happy to provide more details or examples in the revised manuscript if needed.
>
> **[Q3] Table 3 Formatting Issue**
>
> We highlighted the best results among MLLM-based instruction tuning methods. We will clarify this in the revised version of our paper.
>
> **[Q4] Typos & Errors**
>
> Thank you for pointing out our mistakes and we will revise our manuscript for better clarity. FYI, RTPP stands for Relative Text-Position Prediction as specified in line 243, and consists of RPT (Reading Partial Text) and PTP (Predicting Text Position).

---

> > ### Comment · Reviewer_cWFY · 2024-08-13
> >
> > Thank you for your reply. I'm convinced of the value of the method, but the impact is limited by the fact that the code and models are not published as open-source. In today's hyper-competitive environment, this distribution is really essential to have an impact.

---

> > > ### Author Response · Authors · 2024-08-13
> > >
> > > Thank you for your feedback and for recognizing the value of our method. We understand the importance of transparency and reproducibility in research. We have already provided detailed implementation information in our submission and will also include **pseudo-code for the core modules** to enhance transparency and reproducibility further. We are committed to making our work as accessible as possible. Also, if you have any question or additional comments, please do not hesitate to contact us.

---

> ### Author Response · Authors · 2024-08-12
>
> Dear Reviewer cWFY
>
> Because the end of discussion period is approaching, we kindly ask you whether our response is helpful to clarify you or not. Also, if you have any question or additional comments, please do not hesitate to contact us. We thank you for your time and efforts to review our paper.
>
> Best wishes,
>
> Authors.

---

### Author Response · Authors · 2024-08-14
**Summary of Author Responses**

We sincerely thank all reviewers for their constructive and positive comments and we present the summary of our responses to each reviewer as below.

**Concern about ablation study**

While all reviewers agree that our ablation studies are detailed and extensive, reviewer **R-rxzX** and reviewer **R-Pfd9** pointed out that our ablation studies are limited to the moderate variant of our model, BLIP2-OPT. Additionally, reviewer **R-cWFY** raises the question about the effectiveness of the reconstruction loss. To address the concerns about the robustness of our ablation studies, we have conducted **additional experiments on the mPLUG-Owl-based model**, presented in **Table A, C and D**. Due to the limited time constraints and computational resources in the rebuttal period, we focused on components that might be considered most marginal: the reconstruction process and Relative Text-Position Prediction (RTPP) tasks. These experimental results show that the proposed components consistently improve performance on most tasks with both the moderate and large backbone models, BLIP2-OPT and mPLUG-Owl, respectively. We believe these consistent gains demonstrate the effectiveness of our components.

**Analysis of using more scales**

Upon request by reviewer **R-Pfd9**, we present **additional results for the 3-scale configuration** of our model. **Table F** presents the results with varying numbers of scales across each benchmark. We used the BLIP-2-based version of our model and all models are trained using RTPP for the text reading task for this experiment. When integrating the 3rd scale in addition to the 1st and 2nd scale, the number of visual inputs increases about (1+$g$+4$g$+16$g$)/(1+$g$+$4g$) $\\approx$ 4.2 times, while the throughput of our model only decreases by a factor of 2. Note that $g$ represents the number of grids in line 124. The results show that incorporating an additional 3rd scale continues to help with document understanding, but the improvement gain becomes less significant. This is understandable, as the model has already captured most of the key information from the initial two scales, so the additional benefit is reduced. Note that **without our hierarchical feature aggregation module, training the model with all three scales is not feasible** on A100 GPUs due to memory constraints.

**Incremental Novelty (multi-scale features)**

Our main contribution lies not just in utilizing multi-scale features, but in **how we efficiently aggregate them without significantly increasing computational costs**. We present the first practical solution for integrating multi-scale features in high-resolution multi-modal large language models (MLLMs), a challenge previously deemed impractical due to resource constraints. Our framework preserves the advantages of multi-scale features while minimizing computational overhead. Specifically, we developed a hierarchical visual feature aggregation module that reduces computational costs through cross-attentive pooling and a feature pyramid hierarchy, making multi-scale integration computationally feasible for MLLMs. Unlike previous methods that rely on single-scale inputs, our approach enables efficient multi-scale processing, leading to significant performance improvements in MLLMs. During the discussion phase, reviewer **R-CDZQ** agreed that the hierarchical feature aggregation module is a useful contribution to multi-scale feature aggregation.

---

> ### Author Response · Authors · 2024-08-14
> **Summary of Author Responses (2)**
>
> **Comparison to recent SoTA methods**
>
> Reviewer **R-CDZQ** raised questions about our choice of baselines, particularly regarding the recent state-of-the-art (SoTA) methods from leaderboards for each specific task. For reviewer **R-CDZQ**, we have provided a detailed comparison and discussion of these SoTA methods, covering all relevant benchmarks. These SoTA methods typically consist of large foundation models or task-specific methods that may come with external OCR engines. However, it is important to clarify that our primary objective is not to develop a new foundation model from scratch, but rather to enhance existing MLLMs using document data. Additionally, our goal is to build a general document understanding model rather than focusing on specific tasks. Therefore, our comparisons have focused on OCR-free methods that utilize comparable MLLMs in terms of model size and dataset scale, or frameworks pretrained only with document images, such as Donut and Pix2Struct.  Importantly, we demonstrate the effectiveness of our method by successfully enhancing two different foundation models (BLIP-2 and mPLUG-Owl), showcasing the versatility and broad applicability of our approach. We believe that direct comparisons with task-specific models and large foundation models would not fully capture the general-purpose nature of our approach.
>
> Additionally, reviewer **R-rxzX** requested a comparison of our model with two recent concurrent works, TextMonkey and mPLUG-DocOwl 1.5, both of which were released on arXiv in March. We have included these comparisons in **Table B**, using statistics from their papers and official repositories. We adopt mPLUG-Owl as our backbone model whereas TextMonkey and mPLUG-DocOwl 1.5 utilize more advanced Multimodal Large Language Model (MLLM) backbones—Qwen-VL and mPLUG-Owl2, respectively—for initialization. Furthermore, both models benefit from significantly larger training datasets and more trainable parameters, which directly influence their performance. Despite these advantages, our method outperforms TextMonkey in the reported results. Although mPLUG-DocOwl 1.5 performs better than our model, this is likely due to its use of a more advanced backbone and a larger dataset.
>
> We hope that our additional experiments and evaluations have addressed the concerns raised by reviewers and cleared some doubts regarding the proposed method. Thank you.

---

### Decision · Program_Chairs · 2024-09-25

**Decision:**

Accept (poster)

**Comment:**

The manuscript received originally mixed comments. Following the rebuttal and lenghtly discussion between the authors and reviewers, then final assessment overall positive.

The main issue still under discussion, namely the novelty of the use of multi-scale features is clarified, as the contribution clearly lies not in the use of multi-scale features, but in the way of aggregating them in an computationally efficient approach. This is nevertheless something to be clarified in the revised version of the manuscript.

I encourage the authors to properly integrate the clarifications offered during the rebuttal process in the final revision.